



# Identification of runoff formation with two dyes in a mid-latitude mountain headwater

Lukáš Vlček[1], Philipp Schneider[2], Kristýna Falátková[1]

[1]Department of Physical Geography, Charles University, Prague, Czech Republic
5   [2]Department of Geography, University of Zurich, Switzerland

*Correspondence to*: P. Schneider (philipp.schneider@geo.uzh.ch)

**Abstract.** Subsurface flow in Peat Bog areas and its role in the hydrological cycle has garnered increased attention as water scarcity and floods have increased due to a changing climate. In order to further probe the mechanisms in Peat Bog areas and contextualize them at the catchment scale, this experimental study identifies runoff formation at two opposite hillslopes in 10   a peaty mountain headwater; a slope with organic Peat soils and a shallow phreatic zone (0.5 m below surface) and a slope with mineral Podzol soils and no detectable groundwater (>2 m below surface). Similarities and differences in infiltration, percolation, and preferential flowpaths between both hillslopes could be identified by sprinkling experiments with Brilliant Blue and Sodium-Fluorescein. To our knowledge, this is the first time these two dyes have been compared in their ability to stain preferential flowpaths in soils. Dye-stained soil profiles within and downstream of the sprinkling areas were excavated 15   parallel (lateral profiles) and perpendicular (frontal profiles) to the slopes' gradients. That way preferential flow patterns in the soil could be clearly identified. The results show that biomat flow, shallow subsurface flow in the organic topsoil layer, occurred at both hillslopes, however at the Peat Bog hillslope it was significantly more prominent. The dye solutions infiltrated into the soil and continued either as lateral subsurface pipeflow in the case of the Peat Bog, or percolated vertically towards the bedrock in the case of the Podzol. This study provides evidence that subsurface pipeflow, lateral 20   preferential flow along decomposed tree roots or logs in the unsaturated zone, is a major runoff formation process at the Peat Bog hillslope and in the adjacent riparian zone.

## 1 Introduction

Hydrological extremes in Central Europe during recent decades have stimulated debates over sustainable solutions and suitable, cost-efficient strategies to prevent or mitigate the impacts of droughts and floods. Floods on the Vltava and Elbe 25   River are documented back to Middle Ages (Brázdil et al., 2015; Faist, 1967) as having had devastating impacts on infrastructure and economy in large parts of the Czech Republic and Germany. Conventionally, building of dams and reservoirs, and measures to increase catchment-wide retention, are thought to mitigate floods and droughts. For example, a system of dams was built for flood control in the Vltava catchment, the main drainage of the Šumava Mts in the 1960s. However, this system of dams failed to prevent major floods in 2002 (Hladný and Kašpárek, 2005; Hladný, 2009) or 2013.



These floods were mainly caused by heavy rainfall in summer or by rain on snow events. As the Šumava Mts. and the Vltava catchment are promising water resources, research efforts have focused on this area.

Peat Bogs (PB) are prominent in the Šumava Mts., which affects water quality and storm discharge (Ferda et al., 1971; Janský et al., 2008; Vlček et al., 2016). Streamflow in peaty catchments is characterized by its quick rise and fall, and huge volatility: very low baseflows during dry periods and spiky storm hydrographs caused by heavy rainfall events. This behavior has been described in detail in many hydrological studies (e.g. Evans et al., 1999; Holden et al., 2001, 2005); however, most of these studies have focused on pure peat areas only. In the Šumava Mts. Peat Bog covers approx. 35 % of the catchments in this region, but the larger proportion of these catchments is covered by mineral soils. Although Peat Bogs are not dominating the catchment area, their hydrology determines the runoff processes (Vlček et al., 2012). In order to establish the relative roles of Peat Bog and Podzol in this type of catchment, this study focused on a comparison of runoff formation on contrasting soil types – namely a mountain Histosol/Peat Bog (PB) and a Podzol (PZ) on opposite hillslopes of the headwater – representing organic and mineral soil properties, respectively.

In these catchments, retention ability depends mainly on the *shallow depth of the phreatic zone in the Peat Bog*, whereas the *deep phreatic zone in the Podzol* plays a minor role (Vlček et al., 2012). Peat Bog areas are hypothesized to control storm runoff formation in these headwaters. Peat Bogs can significantly contribute to stormflow when the peat is fully saturated, i.e. storm events exceeding a threshold of 10-15 mm (Vlček et al., 2016). According to a geochemical study based on two years of monthly stream sampling data (Kocum et al., 2016), Peat Bogs contribute only 10 % to baseflow at the outlet of the entire Rokytka catchment (3.8 km$^2$). However, some zones of a Peat Bog area, such as springs or soil pipe systems connected to the stream, exhibit high fluctuations in discharge (Holden et al., 2002). This could explain the observed spiky storm hydrographs at the entire Rokytka catchment outlet and at the Rokytka headwater test site (0.6 km$^2$). Pronounced discharge fluctuations from Peat areas could be caused by surface flow (field observation at the Rokytka Peat Bog), near-surface (Holden et al., 2001; 2005) or subsurface stormflow in soil pipes (Rizzuti et al., 2004; Holden et al., 2005; Gerke et al., 2015). Results of Holden and Burt (2003) at a blanket peat site show that near-surface flow (Biomat flow, BMF) up to the depth of 10 cm can contribute more than 90% to the plot's outflow. BMF is defined as lateral stormflow in the organic litter layer with high porosity and high hydraulic conductivity in the topsoil (Gerke et al., 2015).

Sprinkling experiments with sorptive dye tracers have been successfully applied at the plot and hillslope scale in many different environments (Bachmair et al., 2009, 2012; Hümann et al., 201; Flury et al, 1995; Schneider et al., 2014; Weiler and Flühler, 2004; Weiler, 2011; Wienhöfer et al., 2009) and are an established method to identify interactions between infiltration, preferential flow, matrix flow, and percolation. The sorptive dye Brilliant Blue FCF (BB,) is probably the most widely used dye tracer in such sprinkling experiments (Flury et al., 1995). Successful experiments using BB at mineral soil test sites identified significant processes during the formation of storm runoff such as infiltration and initiation of vertical preferential flow in macropores (Weiler and Flühler, 2004), infiltration and vertical preferential flow dependence on soil structures, soil type & land use (Bachmair et al., 2009; Weiler and Flühler, 2004), lateral preferential flow in a soil pipe





network (Anderson et al., 2009; Wienhöfer et al., 2009), and lateral preferential flow in organic topsoil layer/biomat flow (Schneider et al., 2014).

Almost all headwaters of the Šumava Mts., including the Rokytka catchment, have been affected by bark beetle calamity and by storm activity, e.g. the storm Kyrill in 2007 uprooted many trees. These disturbances have been shown to have long-term impact on runoff formation in headwater systems (Langhammer et al., 2015). However, it is unclear how these disturbances modify specific runoff formation processes like infiltration, lateral drainage (soil piping), or percolation in riparian soils.

However, runoff formation at hillslopes, particularly subsurface stormflow (SSF), is highly variable and complex (Bachmair and Weiler, 2012). To evaluate both vertical and lateral preferential subsurface flow in the Rokytka headwater, we conducted two separate sprinkling experiments on each soil type with two different dye tracers with different properties. The experiments in this study were conducted in the vicinity of a headwater stream where the riparian zone connects to the two hillslopes. The test site was selected for several reasons:

1.  The riparian zone represents a potential buffer in hydrologic and hydraulic connectivity between the hillslopes and the stream, thus being a major control of runoff formation (Von Freyberg et al., 2014; Seibert et al., 2009).

2.  Mean groundwater levels at the waterlogged spruce forest, which covers the riparian zone at the Peat Bog hillslope, are deepest at the lowest part of the Peat Bog hillslope PB3 (Fig. 1c). Groundwater is absent at most Podzol soil profiles.

3.  Hydrologic or hydraulic connectivity from hillslopes through the riparian zone to the stream can likely be qualitatively detected and potentially quantitatively described by conservative tracers such as Sodium-Fluorescein (FLC, also called Uranine), e.g. by FLC break-through curves monitored in the stream or at springs. Dye tracer experiments proved to be suitable for answering the question if runoff formation and subsurface flow are dominated by macropore or matrix flow at headwater catchments in the Black Forest (Schneider, 2007).

The sorptive tracer BB was used mainly to detect vertical flow, and the conservative tracer FLC was used to detect vertical and lateral flow at two opposite hillslopes with different soil types (Peat Bog and Podzol). Furthermore, based on other studies in dark-colored organic soils (Markus Weiler, personal communication) we considered that BB-stained soil patterns may be difficult or impossible to identify in the Peat Bog. Thus, we adapted a successful dye tracer soil staining experiment applied in organic forest soils using FLC (Gerke et al., 2008). FLC can be used to identify preferential flow patterns by staining soil particles (Gerke et al., 2015), and as a tracer detecting lateral subsurface flow and thus hydrologic connectivity and potentially tracer breakthrough into a stream or a spring equipped with automated water samplers or field fluorometers. Thus, FLC provides two functions in parallel: vertical and lateral soil staining of preferential flow structures and lateral connectivity between the sprinkling plots (lower, convex part of the hillslopes) and the stream or springs. In theory, such a combined FLC experiment could link plot and hillslope experiments and thus provide additional insights into the mechanical understanding of the entire hillslope-riparian zone-stream system and thus providing an estimate where (flowpaths), when (timing, delay) and possibly how much hillslopes with different soils contribute to discharge in



headwaters. Examples of applications of the dyes BB and FLC are shown in Table 2. To summarize, the objectives of this study are to identify runoff formation at both prevailing soil types (Histosol/Peat, entic Podzol) with emphasis on the following aspects:

- Identify/qualify and estimate/quantify infiltration and vertical preferential flow in the unsaturated zone in soil profiles – as well as possible infiltration barriers causing lateral flow – plot scale,
- Identify soil horizons and/or soil structures, where vertical flow translates into lateral preferential flow, e.g. partially saturated parts in the unsaturated zone vs. expanding fully saturated soil horizons promoting transmissivity feedback and thus hydraulic connectivity – plot scale,
- Identify/qualify and estimate/quantify vertical percolation in the saturated zone – plot scale, identify/qualify if infiltrated water percolates deep into the bedrock and thus a "secondary" (likely strongly delayed) drainage system is recharged during storms – plot scale,
- Identify/qualify lateral preferential flowpaths (hydrologic connectivity) and possibly estimate lateral flow velocities in the soil – hillslope scale.

## 2 Materials and methods

### 2.1 Study site

The Rokytka headwater (3.8 km²) is a tributary to the Otava River, located in the central part of Šumava Mts. The 2nd order stream drains 0.6 km² of the Rokytka headwater covering an altitude range between 1100-1260 m a.s.l. The test site can be divided into two parts – two opposite hillslopes – with different soils and vegetation cover (Fig.1). The mineral soil hillslope consists of a Podzol (PZ hillslope) and is covered by beech stands at the upper hillslope zone PZ1 and dead spruce stands with healthy seedlings at the lower hillslope zone PZ2. The soil profiles are similar throughout the slope without a clear gradient towards the stream. The soil type has been identified as an entic Podzol with a shallow organic top layer (<5 cm) and similar soil texture (Tab.1) to a depth of 1 m. Some small parts of the mineral soil hillslope PZ are covered by haplic Podzol, but these areas are hardly identifiable without excavation. Neither there was a sharp transition between the mineral soil and the bedrock (well-weathered Gneiss or Granite) perceptible with ERT measurements, nor could a persistent groundwater level be detected. The organic soil hillslope is covered by a well-developed mountain Peat Bog (PB hillslope). The entire PB area consists of a mixture of various stages of decomposed peat, however, Acrotelm and lower Catotelm can be distinguished at depths ranging from 8-25 cm. Vegetation and soil depth vary according to the position along the hillslope forming a catena. The upper organic soil hillslope zone is covered mostly by cotton-grass (*Eriophorum L.*) or moss (*Sphagnum L.*) (PB1 in Fig.1). This zone has the highest water table fluctuations and the depth of the PB is 4-5 m. The vegetation cover at the lower hillslope zone (PB2 in Fig.1) consists of pine (*Pinus mungo*), blueberry (*Vaccinium myrtillus*) and moss. The riparian zone (PB3 in Fig.1) forms the bottom of the valley, which is covered by a waterlogged spruce forest





with blueberry and moss. The depth of the PB varies from 1 m in the riparian zone PB3 up to 5 m in the upper hillslope zone PB1 (Fig.1).

Despite the differences between an organic and a mineral soil at the two hillslopes, the basic soil properties – which have a strong impact on infiltration and subsurface stormflow – are rather similar (Table 1). Vertical hydrological conductivity ($HC_v$) was measured on-site with a single-ring infiltrometer (Flow-Group Comp.). Low values of $HC_v$ are contrasted by the high effective porosity. This contrast is caused by relatively few macropores compared to other soils and a high percentage of small pores, which are mostly not active during the infiltration process. A low $HC_v$ in the topsoil is supposed to generate rather surface flow – likely saturation overland flow (SOF) and possibly Hortonian overland flow (HOF) to a minor extent – or near-surface biomat flow (BMF; Sidle, 2007) during high intensity storms. However, at the mineral soil hillslope PZ no surface flow has been observed even during large storms with daily precipitation sums of up to 80 mm. At the organic soil hillslope PB, surface flow can be observed at times when the Peat Bog is saturated.

In general, the dominant runoff formation process in most forested mountain headwater catchments can be described as subsurface stormflow (SSF; Weiler et al., 2006). However, site-specific soil types and their properties, such as those at our test site, Peat Bog (PB3) and entic Podzol (PZ2), may result in a characteristic and possibly unique combination of runoff formation processes. Based on our field surveys and soil mapping using the *Hydrology of Soil Types* classification (HOST, Boorman et al., 1995), the Podzol at the mineral soil hillslope PZ2 was classified as both *hydrological soil class 4*, meaning it is a "*mineral soil, aquifer >2 m depths, no impermeable layer, consolidated substrate*" and *conceptual runoff formation model A*. The latter model implies: "*The dominant water movement is downwards through the vadose zone to an aquifer at least two meters below the surface. Lateral movement is largely confined to the saturated zone, with the hydrological response being controlled by the flow mechanisms of the substrate. Where the rock is more coherent but deeply weathered or fissured, the dominant flow is via the fissures as the bulk of the rock is only slightly porous at best. Aquifers or groundwater are more rarely found in this group*" (Boorman et al., 1995). The Peat Bog at the organic soil hillslope PB3 was classified as *hydrological soil class 12* meaning *organic soils, no significant aquifer, raw peaty topsoil, substrate raw peat, upper soil layers remain saturated for much of the year*, and *conceptual model K*, which implies: "*Where there is deep peat, the flow is dominated by surface and immediate subsurface flow, with the underlying substrate having little influence on the hydrological response*" (Boorman et al., 1995).

**2.2 Hydrological conditions of the Rokytka headwater**

Storm hydrographs at the Rokytka headwater are highly volatile and are characterized by quick and steep rising and falling limbs. The hydrologic response to rainfall events is fast and the recession to antecedent baseflow occurs rather quickly (Fig. 2). The average annual mean flow MQ at the Rokytka headwater outlet is about 0.098 mm h$^{-1}$ (860 mm a$^{-1}$); yet at 330 days of the year (> 90%) the discharge is lower. Compared to the average, the hydrologic year 2015 (01.11. – 31.10.) was a rather dry year with total annual precipitation of 840 mm (long-term average 1220 mm) and total runoff of 580 mm or 0.07 mm h$^{-1}$ (long-term average 860 mm). Mean annual maximum flow MHQ is 2.24 mm h$^{-1}$ and mean annual minimum flow MNQ is





0.04 mm h$^{-1}$. The peak discharge HQ$_p$ in 2015 reached 3.5 mm h$^{-1}$ (Fig. 2), thus the ratio HQ:MQ ≈ 50 is relatively high. From June to October 2015 daily precipitation rarely exceeded 10 mm d$^{-1}$ and thus stormflow events were unfrequent and small. Yet, the Rokytka creek did not fall dry due to two persistent springs at the mineral soil hillslope PZ. In contrast, springs at the organic soil hillslope PB are susceptible to desiccation.

## 2.3 Dye tracer experiments

The dye tracer experiments were carried out at both hillslopes (mineral soil slope PZ2 and organic soil slope PB3) of the Rokytka headwater during baseflow conditions in late June 2015. At each hillslope two 1.5 m x 1.5 m plots were sprinkled with both dyes (Brilliant Blue, CAS#3844-45-9; Sodium-Fluorescein, CAS#518-47-8). All sprinkling plots are located at the transition between the concave, lower part of the hillslope and riparian zone in the vicinity of the stream (distance to stream ≈ 10m; Fig. 3).

First, all plots were pre-sprinkled with 45 L (≈ 20 mm) of local stream water to raise soil moisture and connect the pathways for water percolation, and then the plots were sprinkled with 45 L (≈ 20 mm) of dye solution. The overall sprinkling time at each plot was ~ two hours in order to simulate a rainfall intensity of 20 mm h$^{-1}$. These amounts and intensities represent a heavy rainfall storm in the Šumava Mts.. Due to previous rainfall events, the soil moisture ranged between 0.40-0.45 VWC. A 40 mm rainfall usually causes significant stormflow and also represents frequently occurring amounts of daily precipitation in Central Europe low mountain ranges (Hümann et al., 2011). Storms of this magnitude occur 2-3 times in an average year in the Šumava Mts. (Fig.2). The groundwater level in the Peat Bog was initially about 0.35 m below the terrain surface (Fig.2), which represents average Peat Bogs summer groundwater levels at the near-riparian organic soil hillslope.

PH is an important parameter when using FLC as a soil staining dye for preferential flow identification (Gerke et al., 2008, 2013). The soils in the Šumava Mts. are characterized by low pH values ranging from extremely acidic pH 3.8 for Peat Bogs to moderately acidic pH 6.0 for Cambisols. In the Rokytka headwater soil pH is very strongly acidic pH 4.8 at the Peat Bog plot PB3 and strongly acidic pH 5.4 at the mineral soil plot PZ2. Therefore, a NaOH solution was added during the initial pre-sprinkling with dye-free water (pH 12) to raise and buffer the soil pH to reduce fluorescence suppression of FLC caused by a very strongly acidic environment.

The experiment continued with excavation of the FLC sprinkling plots. The excavation of soil profiles and the photography of FLC-stained soil structures were performed under short-time UV illumination (410 nm) at night, approx. 4 hours after sprinkling, as FLC is strongly light sensitive (Käss, 1998).

In the following we describe the soil profile excavation and photo documentation procedure performed after the sprinkling step by step. First, we visually surveyed the terrain surface and all micro-depressions downslope of the sprinkling plots along the primary topographic gradient (thalweg) towards the stream to identify potential preferential flow patterns near or at the surface. In case of visible dye patterns at the surface, exploratory frontal soil pits, perpendicular to the horizontal/lateral flow, would start at the stained surfaces nearest to the stream. Second, the systematic frontal profile





excavation started 1.5 m downslope from the sprinkling plots outside of the sprinkling area along the projected flow direction along the thalweg (Fig. 4). Third, the pre-planned systematic excavation was extended along possible secondary gradients using additional exploratory frontal profiles at local terrain depressions. These were excavated up to 10 m down the slope to find potential secondary flow directions and to identify the maximum distance of dye-stained flow structures.

Fourth, lateral profiles, oriented parallel to the flow, were excavated in a systematic way at all places where the dyes were detected outside of the sprinkling area. Fifth, the sprinkling area was excavated in a pre-defined systematic way (Fig. 4). The pictures of soil profiles were taken from two sides in frontal and lateral orientation. Frontal images were taken towards the slope (perpendicular to the direction of horizontal-lateral flow), whereas lateral images were taken along the slope (parallel to the direction of horizontal-lateral flow). The size of each image (photograph) was 50 cm x 50 cm. Frontal pictures were

taken in soil pit profile rows at every 0.25 m (Fig. 4a), lateral pictures (Fig. 4b) in soil pit profile lines every 0.5 m (G – L). A similar system of excavation was used by Schneider et al. (2014) and Gerke et al. (2015).

Pictures of the soil profiles were taken during the excavation with a digital Micro Four Third camera with a crop factor of 2.0 (Panasonic Lumix DMC-G1 with a 12 MP MOS sensor, 13 mm x 17.3 mm sensor area, and a 14-45 mm zoom lens, mostly at a focal length of 14 mm, equivalent to 28 mm in full format/35 mm film) under daylight conditions beneath

a shading tarp to avoid direct sunlight and shadow effects in case of the BB plots. White balance, white point and black point reference were calibrated using a Datacolor Spider Cube, which was placed in every image. Pictures at the FLC plot were taken at night with the same camera. Each FLC soil profile was illuminated separately with two light sources:

1. A 500 W Halogen lamp (approx. light temperature 5500 K in the visible spectra with maximum 550 nm) to document the natural soil profile color with its horizons;

2. A 27 W UV LED lamp (9x 3W UV LEDs, Highlite International BV Comp.) producing UV light 410 nm with a 120° beam angle to visualize fluorescent FLC stained soil structures similar to Gerke et al. (2013).

The dye-stained flow patterns for both dyes BB and FLC at all soil profiles were analyzed according to a method and with an analytical tool described by Weiler and Flühler (2004). This method was originally developed for analyzing BB. Therefore, the color space of the photographs is converted from the Red-Green-Blue (RGB) color space taken by the camera

sensor into the Hue-Saturation-Value (HSV) color space and then classified and spatially analyzed with an algorithm written in IDL code (Weiler and Naef, 2003). For the Rokytka experiments, this procedure was applied for both dyes, BB and FLC, thus for two different groups of photographs. To detect and analyze FLC in the soil profile photographs similarly to the BB photographs, the dye detection routine in the original IDL code was adapted for optimal FLC identification.

## 2.4 Headwater stream and spring sensing and sampling

All sprinkling plots were located approx. 10 m away from the headwater stream in the lower concave part of the hillslope. Rainfall runoff data at the Rokytka catchment indicate that 10-15 mm of precipitation produce a noticeable response in the stream when the Peat Bog is in moist, near-saturated conditions (Vlček et al., 2016). Hence, the amount of sprinkling water (40 mm per plot) simulated a rainfall that connects the hillslopes to the riparian zone and the headwater stream. Thus, the



tracer FLC could potentially appear in the stream. During early summer conditions, similar to the conditions of the experiments with comparable antecedent soil moisture and groundwater levels, 40 mm of daily rainfall (on the entire catchment) rises the discharge from baseflow to peak flows on the order of 1.5 mm h$^{-1}$. To detect whether and when the hillslopes connect to the stream, we installed two ISCO samplers in the vicinity of the sprinkling area in the stream and one

at the catchment outflow to sample stream water for future analysis (Fig. 1). Each sampler detected one hillslope. A field fluorometer (Albilia GGUN-FL30, detection limit 2 x 10$^{-11}$ g mL$^{-1}$) was installed in the stream at the Rokytka headwater gauge in order to detect hydrological connectivity, to capture the travel time of FLC from the sprinkling plots to the stream, and to possibly monitor the FLC break-through curve. The sampling interval was 15 min, allowing for continuous operation of the field fluorometer two weeks.

**3 Results**

**3.1 Mineral soil hillslope, Podzol (PZ2)**

**3.1.1 Brilliant Blue (BB)**

Visible dye-stained patterns of lateral preferential flow in the soil profiles were observed up to a distance of 0.5 m outside the BB sprinkling plot in the downslope direction (± along the thalweg). Additional exploratory trenches were excavated two

and three m downslope from the plot to detect further dye-stained patterns of lateral subsurface flow in the soil, but no traces of BB were found. Thus, the systematic excavation started 1.5 m downslope of the BB sprinkling plot (Fig. 4). Frontal profiles (orthogonal to the slope) are depicted in Fig. 4a and lateral profiles (parallel to the slope) are depicted in Fig. 4b.

Fig. 5a shows a selected lateral soil pit profile (IL1), which is mostly within the BB sprinkling plot at the mineral soil hillslope. BB infiltrated rather homogeneously into the upper soil horizon (O+A) and percolated rather heterogeneously

deeper into the soil. Thus, BB-stained patterns are placed irregularly at lower soil horizons, some of which reached down to the B/C horizon without continuous connection to the topsoil in the excavation plain.

The frontal soil profile image (Fig. 5b) confirms the prominence of BB-stained lateral-horizontal flowpaths in the shallow subsurface, namely in the uppermost soil horizons (O+A, A/B). Some BB-stained patterns were observed at deeper soil horizons of the profile indicating preferential infiltration independent of specific soil horizon or depth. Stones and roots

occur rarely and thus do not significantly modify lateral or vertical dye-stained flow structures. BB was transported vertically along patches in the soil matrix, typically at locations where the higher soil horizons O+A and/or A/B were stained. These vertical preferential flows were rather created by slight differences in texture and porosity of the soil matrix than by vertical macropore structures such as root networks or burrows. There were no visible vertical macropores. The root systems of the spruce stands are mostly limited to a depth of 0.2 m (O+A and A/B horizons) and traces of edaphone are very limited. As

this frontal soil profile AC0.5 (Fig. 5b) is located outside the sprinkling plot, the local soil was not pre-sprinkled and thus likely less saturated when compared with the lateral profile located within the sprinkling plot (Fig. 5a). This may explain the



more pronounced dye-stained vertical flow structures in the frontal soil profile AC0.5 compared to the pre-sprinkled lateral profile IL1.

### 3.1.2 Fluorescein (FLC)

The dye FLC was not visible at the soil surface outside of the sprinkling plot. First exploratory trenches were excavated at a distance of 2 m downslope of the sprinkling plot; none of them showed any dye-stained patterns in the soil profiles. The systematic excavation started 1.5 m downslope from the sprinkling plot similar to the BB plot at hillslope PZ2 (Fig. 1). FLC dye-stained patterns are located without any visible link to the soil horizons, roots or stones (Fig. 6a and 6b). FLC dye was sparsely distributed in the topsoil horizons (O+A, A/B). Almost no dye-stained patterns were found in the lateral soil profiles, especially in the lower soil horizons, within the sprinkling plot area (e.g. Fig. 6a). The largest occurrence of FLC dye-stained soil patterns were found at frontal profile AC0.5 in the Bh horizon (Fig. 6b).

The smallest stained spots (<1 cm²) were likely caused by UV light reflected from small grains of quartz from the weathered Gneiss. The adopted version of the FLC dye classification algorithm could not distinguish these pixel-scale patches from truly FLC dye-stained features. However, this analytical bias has a negligible impact on the main findings concerning the FLC dye-staining in the mineral soil profiles (PZ2).

## 3.2 Organic soil hillslope, Peat Bog (PB3)

### 3.2.1 Brilliant Blue (BB)

The visual survey of the soil surface in the vicinity of the BB sprinkling plot revealed near-surface flow in the NW direction towards the stream. BB was detected in a small, water-filled depression 10.5 m downslope from the sprinkling plot. Excavation was performed from this point uphill towards the dye sprinkling plot (Fig. 4a, yellow section), following the dye-stained soil patterns by excavating and photographing frontal soil profiles every 0.5 m. The BB stained flowpath did not strictly follow the terrain gradient but went from the NW side of the sprinkling plot and followed mostly lateral preferential flow structures formed by decomposed trees or roots. This lateral preferential flowpath was later identified as the main direction of the subsurface flow. Relatively smaller and less stained flowpaths were observed downslope from the sprinkling plot (Fig. 4a, orange section), with BB disappearing two meters from the sprinkling plot. The BB excavation started the day after the sprinkling, yet dye stored in large macropore pockets started flowing down the trench walls when these soil structures were truncated.

BB followed lateral soil pipes that were formed by decomposed roots or fallen trees. Healthy trees and undecomposed timber did not create such effective lateral preferential flowpaths, therefore, they had no significant impact on dye-stained patterns (Fig. 7). BB created clearly detectable dye-stained patterns on the dark peat particles as well, so the major flowpaths of BB could be detected even several days after the dye application.



The excavation of BB stained soil patterns at the organic soil hillslope PB3 proceeded from two directions (NW and SW, Fig. 4a) following the stained flowpaths in the soil. Near the sprinkling plot, most of the dye was detected at the surface and in near-surface soil horizons, which correlates with Acrotelm (Fig. 7). About two meters downslope from the BB sprinkling plot at hillslope PB3 (Fig. 1) the dye-stained patterns diminished in the Acrotelm and were observed mainly in and around

macropores in the Catotelm. The excavation caused problems at location FD1.75/profile D $0 - 1$ as dye-filled macroporous pockets in the soil drained BB when disturbed during excavation and thus secondarily stained these soil profiles. Such secondary patterns were cleaned to minimize falsely detected BB along the excavation front.

### 3.2.2 Fluorescein (FLC) in the soil profiles

The excavation of the FLC sprinkling plot at PB3 at the organic soil hillslope (Peat Bog) started 3 hours after sprinkling after

dark. Repeated visual surveys of the terrain surface and of the excavated soil trenches downslope of the sprinkling plot using the UV-lamp (410 nm) and UV torches (385 nm) did not detect any traces of FLC at the surface or in the soil pits. Even in the soil horizons within the sprinkling plot, no FLC dye-stained flow patterns were identified. Only parts of the vegetation at the surface of the sprinkling plot itself were visibly stained with FLC. FLC was later detected in a small water-filled depression about 2.5 m downslope from the sprinkling plot PB3.

### 3.3 Fluorescein (FLC) in springs and in the stream

Traces of FLC were not detected in the headwater stream, neither in any water samples taken by automated water samplers (ISCO 6700) or water-level proportional water samplers WLPWS (Schneider et al., 2013), nor with the field fluorometer during the 14-day monitoring period following the dye application. The fluorometer (Albilia GGUN-FL30, detection limit 2 x $10^{-11}$ g mL$^{-1}$) was installed at the headwater outlet next to the gauge to possibly detect either hydrologic connectivity in the

soil during or shortly after the sprinkling experiments or hydraulic connectivity via a groundwater flow system with some time delay. The auto-samplers monitored the two main streams at their confluence, whereas the WLPWS were installed at the springs and as a backup at the gauging station. Unfortunately, none of the in-stream or at-spring devices detected any FLC.

### 4. Discussion

Hillslope hydrology is concerned with the partition of precipitation as it passes through the vegetation and soil between overland flow and subsurface flow (Kirkby, 1988). Runoff formation at hillslopes (zero-order basin response) and in riparian zones (1st and 2nd order stream response) are the main controls defining the hydrological response of mountainous headwaters in humid climates. Vertical processes such as infiltration, (preferential) percolation, and deep percolation into the bedrock together with lateral connectivity between hillslopes, riparian zone, and headwater streams are 1st order controls,

which determine storm runoff and retention at mountain headwaters. Lateral connectivity between hillslopes and streams can



occur either as *hydrologic connectivity* at the soil surface, in the vadose zone or as *hydraulic connectivity* in the phreatic zone. Both, vertical processes and lateral connectivity are strongly dependent on soil properties, thus soil types are a major regulator of the interplay of these processes. Conversely, the spatial distribution of soil types is partly dependent on topographic location (Catena concept) as well as on proximity to drainage. Consequently, soil types can be classified in hydrologically meaningful terms, e.g. according to HOST (Boorman et al., 1998). Dye tracer sprinkling experiments with Brilliant Blue (BB) and Fluorescein (FLC) at two opposite hillslopes with hydrologically contrasting soil types – namely a Peat Bog and a Podzol hillslope – allowed us to to test our runoff formation hypothesis at the Rokytka headwater (0.6 km$^2$) and the Rokytka catchment (3.8 km$^2$).

Hillslope processes define how small catchments respond to rainfall (Anderson and Burt, 1990). Specifically, hillslope processes control how long water is stored in soil or bedrock, which determines how quickly small catchments respond to rainfall (Uhlenbrook et al., 2008). Our experiments in the Šumava Mts. showed that the Peat Bog hillslope connected much more quickly to the stream and contributed considerably more to the runoff response of the headwater than the Podzol hillslope. The larger Rokytka catchment (3.8 km$^2$, 3$^{rd}$ order stream) showed similar hydrological behavior (Fig. 2) – low baseflow and flashy storm hydrographs – to the smaller Rokytka headwater (0.6 km$^2$, 2$^{nd}$ order stream). This is noteworthy since the proportion of the Peat Bog ranges from 60% at the 2$^{nd}$ order stream headwater to less than 30% at the 3$^{rd}$ order stream catchment; the remaining areas are covered by Podzol. This illustrates that the hydrologic response of the catchment is dominated by the runoff formation at the Peat Bog, whereas we speculate that a deep groundwater system is fed by percolation in the Podzol that is rather disconnected from the 2$^{nd}$ and 3$^{rd}$ order streams.

## 4.1 Mineral soil hillslope, Podzol (PZ2)

### 4.1.1 Brilliant Blue (BB)

Based on the properties of Podzol, surface flow (SOF or HOF) at the mineral soil hillslope PZ was not expected. This hypothesis was supported by the results of the BB staining patterns in the soil profiles at PZ2. However, the dye did not identify a hydrologic active soil horizon as clearly as with the Gleysol hillslope study (Schneider et al., 2014). The most abundantly stained soil structures (volume density of up to 85%, Fig. 5b) were found within the uppermost soil horizon O+A, up to a depth of 0.1 m below the surface. The shape of the BB depth distribution was similar at the plot, however, the volume density of max. 50% is significantly lower (Fig. 5a).

Based on the soil properties at the mineral soil hillslope PZ2, such as porosity and hydraulic conductivity, infiltration was expected to be rather stable and homogeneous. However, BB infiltrated heterogeneously. Parts of the topsoil created conditions for the occurrence of fingering (DiCarlo et al., 2013) or similar types of matrix preferential flow (Weiler et al., 2004; Anderson et al., 2009; Wienhöfer et al., 2009). Similar to previous work, we attribute the heterogeneous infiltration to the corrugated transition between the dark organic topsoil horizons (O, A) and the lower mineral horizons (A/B, B).





However, Weiler (2004), Anderson (2009), and Wienhöfer (2009) conducted their studies at steeper slopes and on different soil types compared to the Šumava experiments.

A sharp interface between an upper organic and a lower 'organo-mineral' layer like in the Šumava sprinkling experiments can initiate significant biomat flow (Gerke et al., 2015) that can be attributed to water repellency (Doer et al., 2000). Water repellency of the soil surface was observed in the organic topsoils at the Rokytka headwater during dry periods. However, due to rainfall events prior to our experiments, the antecedent soils moisture conditions (0.45 − 0.5) likely did not support water repellency in the topsoil (Fig. 2).

At PZ2, BB was clearly visible at deeper soil horizons, where it created seemingly detached stained patterns. These stained patterns represent vertical preferential flowpaths (Nobles et al., 2010; Gerke et al., 2015; Uchida et al., 2005), but the excavation spacing (0.25 − 0.5 m) was probably too coarse to detect connected stained flowpaths in full detail. Horizontal-lateral preferential flowpaths dominated at the topsoil layers (O+A), whereas vertical flow directions (percolation) dominated at the lower soil horizons. These differences in the direction of stained pathways are consistent with the results of Schneider et al. (2014) or Gerke et al. (2015).

### 4.1.2 Fluorescein (FLC)

FLC stained significantly fewer soil structures or pathways compared to BB at both hillslopes. FLC was almost absent with few exceptions in the near-surface organic topsoil layers. This could be because of the relatively small amount of dyed water (20 mm dyed sprinkling water and 20 mm undyed pre-sprinkling water). Previous FLC soil staining experiments used simulated rainfalls of 50-100 mm (Gerke et al., 2015). The size of the irrigation plots (1.5 m x 1.5 m) does not appear to be a factor, as previous work used irrigation plots that were 1 m x 1 m, and successfully detected FLC (Gerke et al., 2015). The FLC dye solution could have bypassed the topsoil horizons via macropores and soil pipes without visibly staining these preferential flow structures due to various causes, e.g. local hydrophobicity or strong acidity. On the other hand, FLC stained patterns in the lower soil horizons at the Podzol plots are similar to the BB stained patterns. This indicates, that the organic topsoils at the Šumava test sites may suppress the fluorescence of FLC in addition to the well-known pH induced fluorescence elimination. At the Podzol hillslope PZ2 the FLC stained soil patterns suggest rather a subsurface lateral pipeflow network as described in Uchida et al. (2005) than biomat flow as identified by Gerke et al. (2015).

### 4.2 Organic soil hillslope, Peat Bog (PB3)

### 4.2.1 Brilliant Blue (BB)

BB at the Peat Bog hillslope PB3 did not detect surface flow SOF or significant vertical deep percolation. BB patterns in the soil profiles supported the hypothesis of surface-near biomat flow, which can be attributed to lateral preferential flow in the Acrotelm. Differences in porosity and hydrological conductivity between Acrotelm and Catotelm create similar stained



patterns when compared to the Gleysol hillslope studied in the experiment of Schneider et al. (2014). The infiltration process at the hillslope PB3 matches fairly well the definition of lateral subsurface stormflow (Wienhöfer et al., 2009).

Lateral preferential flow was detected with BB at PB3, however, it was limited mostly to a few, but well-connected pipe networks with high drainage capacity. These soil pipe networks are created by decomposed dead trees or dead roots in the Acro- and Catotelm. Such lateral soil pipe networks are an important runoff formation process at peaty catchments (Jones, 1997; Holden et al., 2002). Holden and Burts (2003) identified the dominant lateral stormflow as shallow subsurface flow (SSF) down to the depth of 0.1 m. However, BB at the organic soil hillslope PB3 showed that lateral soil pipes were connected both in Acro- and Catotelm (depth 0.1-0.4 m) to jointly form the major preferential flowpaths through the Peat Bog. Hence, during the BB excavation soil pipes were only observed in the Catotelm. BB dye-stained flowpaths in the soil appeared prominently further downslope (outside of the dye sprinkling plot) and connected mostly laterally via soil pipes rather than vertically penetrating the Acrotelm. However, most soil profiles – both frontal and lateral – document that the BB dye-stained flowpaths are rather limited to macropore structures and rarely to matrix flow. Deep percolation at the organic soil hillslope PB3 was not detected. The assumption that BB may be difficult to optically detect in such dark soils as Peat Bog, was not confirmed. It could be shown that BB can be successfully applied in Peat Bogs to stain vertical flow structures in soil profiles and has the potential to trace rather long-distance lateral preferential flowpaths (distances > 10 m) in waterlogged areas with shallow groundwater (~ 0.5 m below surface).

### 4.2.2 Fluorescein (FLC)

The critical role of pH affecting FLC's fluorescence and thus its on-site optical detection is well known (Gerke et al., 2013; Käss, 1998). Therefore, the FLC solution was buffered with NaOH to compensate for the strong acidity of the organic soil, similar to the approach successfully applied in FLC staining experiments in organic topsoils in Japan (Gerke et al., 2015).

The rarely detected FLC at the Peat Bog hillslope does not necessarily indicate that no dye infiltrated into the upper organic layer as the dye's fluorescence could not be optically detected in-situ. The FLC solution could have bypassed the organic topsoil horizons via macropores and soil pipes at the Peat Bog plot PB3 as well as at the Podzol plot PZ2. The very low soil pH in the Peat Bog might be the reason why FLC fluorescence was not observed. In solution, FLC displays maximum fluorescence at pH 8.5 and by pH 5, fluorescence is no longer observed (Käss, 1998). In soils, the pH limit for detection of fluorescence ranges from pH 5 to 6. (Gerke et al., 2013). Although we attempted to increase the soil water pH by pre-sprinkling the plots with NaOH-enriched water (pH 12), and similarly buffered the FLC solution, these countermeasures at the sprinkling plots were probably not enough to significantly change the pH conditions in the soil at both hillslopes, but especially at PB3.

FLC was mostly visible on organic surfaces such as plants (moss, grass) on the surface but not in the peat itself. This might be attributed to the tendency of FLC to attach to organic matter, which significantly counteracts its conservative behavior in mineral soils, especially in thick organic soils such as Peat Bogs. Furthermore, compared to the conditions of the successful organic topsoil staining in Japan (Gerke et al., 2015), the FLC fluorescence in PB3 might be affected by the low-





pH water in the phreatic zone (~ 0.5 m below surface) and in the capillary fringe. The "very strong acidity" in the Peat Bog soil, the phreatic zone, and the capillary fringe together may have diluted any buffering effect of the NaOH enriched sprinkling water. However, although not detected during the night, some FLC was visible the next day during daylight in a small, water-filled depression approximately three meters downstream from the FLC sprinkling plot. This means that the dye must have been transferred via preferential flow in lateral soil pipes in the vadose zone, and not been either affected by pH changes or irreversibly bound to the organic soil particles.

## 4.3 Plots-stream hydrological connectivity with Fluorescein (FLC)

Limitation and possible causes of failure of FLC staining and tracing experiments have been well described for organic and mineral soils by Gerke et al. (2008) and for ground- and surface water applications by Käss (1998). The most common problems were addressed for the Šumava experiments by:

- Pre-sprinkling of the plot with NaOH buffered water (pH 12);
- Addition of NaOH to the FLC solution (pH 12) to create a secondary pH buffer;
- Avoiding bright sunlight as FLC is light-sensitive and decays quickly, by performing the FLC sprinkling and excavation work at night with controlled short-time exposure to UV and visual spectra light.

Despite our advanced and dense FLC monitoring network in all tributaries and springs draining the hillslopes field fluorometer, automated water-samplers and water-level proportional water samplers (Schneider et al., 2013), FLC was not detected in the stream or at the springs. As a result, we could neither measure a tracer breakthrough nor could we delineate a transit time or prove hydrologic connectivity from the sprinkling plots to the drainage system. The following explanations may explain why FLC did not reach the drainage system:

- Small sprinkling water volume and limited area of the sprinkling plots (1.5 m x 1.5m) compared to the distance from the plots to the stream or springs (10 m);
- Masking of the FLC fluorescence by very low background pH values (Käss, 1998);
- Fluorescence reduction and masking of FLC by organic substances (Käss, 1998);
- Sorption of FLC to vegetation and topsoil organic matter.

As the sprinkling solution mixed with soil and shallow groundwater, its pH probably dropped back to low background pH values. Another reason for null detection of FLC might be the relatively small amount of water sprinkled at the plot compared to the soil and groundwater volume. The influence of the capillary fringe and the soil matrix to dilute the pH enriched sprinkling water might be significant, but the presented data cannot confirm or disprove this possibility.

## 4.4. Dominant runoff formation

In extension to our hypothesis, which was based on the conceptual runoff formation model, HOST model A, (Boorman et al., 1995; Fig. 1 and Fig. 8) we found that additional lateral subsurface flow in deeper soil horizons occurred at the mineral





hillslope PZ2 (Podzol). As predicted, a surface-near runoff formation process occurred in parallel to deep percolation (Fig. 8). Biomat flow, a shallow subsurface flow in the topmost soil horizon O+A (Sidle et al., 2007; Gerke et al., 2015) is thus a relevant runoff formation process at the mineral hillslope PZ2 (Podzol). The absence of a shallow groundwater body and percolation-restricting soil layers buffers the lateral stormflow at the hillslope PZ2, as a large portion of precipitation

infiltrates into the soil and percolates into the underlying bedrock. These findings are supported by the lack of temporary or fully saturated zones, perched aquifers or a groundwater body rising into the soil (transmissivity feedback), which may connect the hillslope to the stream and thus effectively drained the Podzol (Fig. 8). According to the runoff formation decision scheme by Scherrer and Naef (2003), the dominant runoff formation process at the Podzol hillslope can be classified as a combination of delayed Hortonian overland flow (HOF) and delayed subsurface stormflow (SSF2). The

sprinkling experiments with the dyes (BB and FLC) showed the existence of vertical and lateral preferential pathways in the topsoil and indicated that delayed subsurface flow SSF occurs also.

At the organic soil hillslope PB3, the hypothesized HOST model K (Boorman et al., 1995) was supported by our results. Biomat flow and lateral preferential flow in soil pipe networks formed by decaying fallen trees and roots in the Acrotelm and – in addition to our prediction also in the Catotelm – are the primary runoff formation processes at the Peat Bog hillslope

(Fig. 8). Our hypothesis of HOF was confirmed for the Peat Bog hillslope PB3 by the results of the sprinkling experiments with Brilliant Blue.

## 5. Conclusion

In this study runoff formation during stormflow was investigated at two opposite hillslopes with different soil types (Histosol/Peat, Podzol) in a 2nd order mountain headwater catchment in the Czech Republic. Two dye tracers with different

attributes – the sorptive dye Brilliant Blue (BB), and the conservative dye Fluorescein (FLC) – were applied to stain preferential flowpaths in the soil and at its surface. Runoff formation processes were identified by dye-stained preferential flow structures mostly in the unsaturated zone. These results illustrated the importance of lateral and vertical preferential flowpaths at both hillslopes.

At the Peat Bog hillslope, BB staining identified a quickly activated and effective shallow lateral subsurface drainage

system in the Acrotelm. Preferential flow structures connected the hillslope with the not-sprinkled riparian zone via lateral pipeflow along decayed roots and fallen trees in the Acrotelm, and the upper Catotelm. However, the dye was not observed to have reached the stream. Healthy roots do not create similar drainage-effective, well-connected lateral preferential flow structures as decomposed roots or dead trees at the Peat Bog site. In contrast, subsurface flow at the Podzol hillslope was created only near surface in the organic topsoil (biomat flow). The lateral subsurface transport in the unsaturated zone at the

organic soil hillslope (Peat Bog) was about an order of magnitude higher (10 m lateral flow) than at the mineral soil hillslope (Podzol, 1 m lateral flow).




We were more easily able to detect hydrologic connectivity from the hillslope to the riparian zone with BB than with FLC at the Peat Bog hillslope. This is surprising, as BB is considered to be more sorptive (less conservative) than FLC. Moreover, the dark-blue BB is often difficult to optically detect in dark organic soils like Peat. We attribute this finding to the fact that BB is less affected than FLC by the very strongly acidic soil and groundwater (pH <5) found in peaty

environments such as the Šumava headwaters. Although NaOH was added to the pre-sprinkling water and the sprinkled FLC solution to compensate for FLC's sensitivity to strongly acidic environments, FLC was not detected in the topsoil at either hillslope and showed limited detection in the subsoil, especially at the Peat Bog. In addition, the determination of preferential flow paths with dye tracers relies on strong sorption of the dye to organic matter in the soil. The relatively conservative dye FLC exhibits this property, however in this case, we hypothesize that FLC adsorbed to organic surfaces,

reducing the optical detectability of the dye's fluorescence at the Peat Bog hillslope.

At the mineral soil hillslope (Podzol) both dye tracers worked well in the subsoil and delivered similar results. Percolation in the soil and deep percolation into the bedrock dominated, as expected; lateral preferential drainage was rather limited. Compared to the organic soil hillslope (Peat Bog), the lateral subsurface flow distance is reduced by an order of magnitude (1 m vs 10 m). The findings at the Podzol hillslope with prevailing vertical flow agree with the facts that

groundwater influenced soil horizons could not be detected in soil profiles or with an ERT survey (no saturated zone in depths up to 2.5 m). The absence of a phreatic zone in soil profiles in the Podzol hillslope contrast with the persistent shallow saturated zone (<0.5 m) at the Peat Bog hillslope, which is reflected in the different runoff formation processes, preferential flow structures, and lateral drainage at the Rokytka headwater.

*Author contribution:* L. Vlček, K. Falátková, and P. Schneider participated in the experiment and contributed to the

manuscript.

*Competing interests.* The authors declare that they have no conflict of interest.

*Acknowledgements.* The authors would like to thank Jan Seibert from the University of Zürich for providing equipment and tools and Markus Weiler from the University of Freiburg for providing his IDL code for the Brilliant Blue image analysis. We are indebted to Bohumír Janský, Václav Královec, Miroslav Šobr, Michal Jeníček, Zdeněk Kliment, Luděk Šefrna, Jan

Kocum, and Julius Česák from the Charles University, Prague, for support and assisting with fieldwork and to the National Park Šumava for permission to conduct our research on its land. This work was supported by grant GAČR 13-32133S "Headwaters retention potential with respect to hydrological extremes," and by grant COST ES1306, LD 15130 "Impact of landscape disturbance on stream and basin connectivity".




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





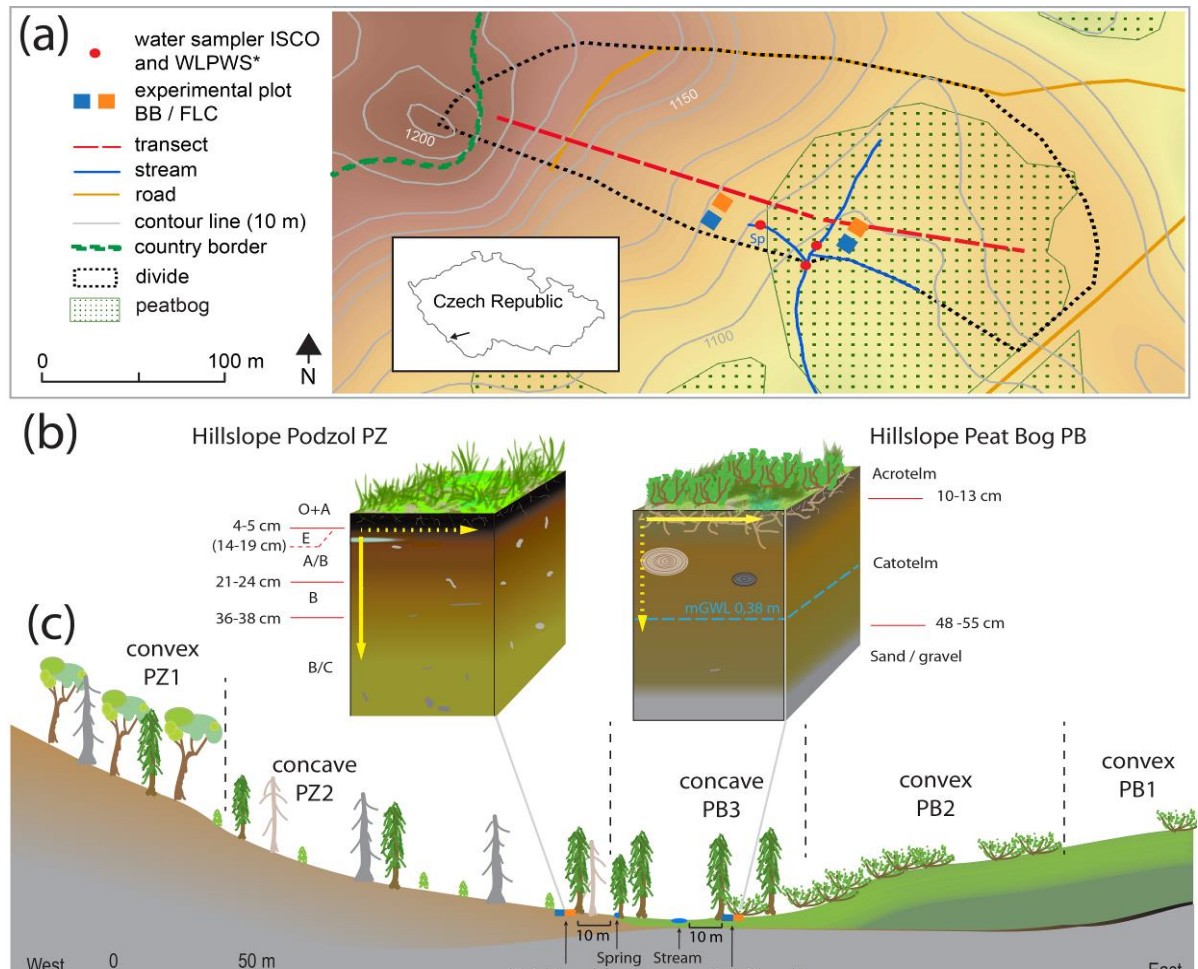

**Fig. 1. (a)** Overview of the Rokytka headwater test site (0.6 km²) in the Šumava Mts. in SW Czech Republic; Sp = spring. **(b)** The yellow arrows at the experimental Podzol plot PZ2 and the Peat Bog plot PB3 represent simplified conceptual models of assumed runoff formation during stormflow according to the *Hydrology of Soil Types* classification (HOST; Boorman et al., 1995): The soil type entic Podzol at plot PZ2 matches HOST soil class 4 with conceptual model A; the soil type Peat Bog at plot PB3 matches HOST soil class 12 with conceptual model K (Fig. 8). **(c)** Cross-section of the Rokytka headwater and its two opposite hillslopes. The mineral soil hillslope Podzol (PZ) consists of entic or at least haplic Podzol and is covered by beech and spruce stands. The organic soil hillslope (PB1-PB3) represents a typical Peat Bog (PB) of the Šumava Mts. The upper hillslope zone PB1 (cotton grass), the lower hillslope zone PB2 (pine), and the riparian zone PB3 (waterlogged spruce forest) represent zones of the PB catena with different vegetation cover, groundwater, and peat soil depths.





**Table 1.** Soil characteristics at the two experimental plots PZ and PB in the Rokytka headwater. PZ = Podzol (mineral soil), PB = Peat
Bog (organic soil); RH = Coverage of a selected soil type at the Rokytka Headwater; OR = Coverage of a selected soil type at the Otava
River catchment; *Peat in general – Histosol according to IUSS Working Group WRB (2006); ** Depth at hillslope PB3; SL = Sandy-
loam; L = Loam.

| Slope | Soil type | Soil area RH / OR [%] | Soil depth RH / OR [m] | Depth to groundwater [m] | Slope [%] | Soil layer | Eff. porosity [%] | Hydrol. cond. [mm h⁻¹] | Soil texture |
|---|---|---|---|---|---|---|---|---|---|
| PZ | entic Podzol | 51 / 45 | 0.7 / 0.4-0.7 | > 2.0 | 4-6 | A/B | 80.0 | 0.50 | SL-L |
|    |              |         |               |       |     | B   | 53.5 | 0.38 | SL |
| PB | Histosol* | 44 / 20 | 2.1 / 1.2** | 0.35 | 3-5 | T | 89.2 | 0.30 | - |




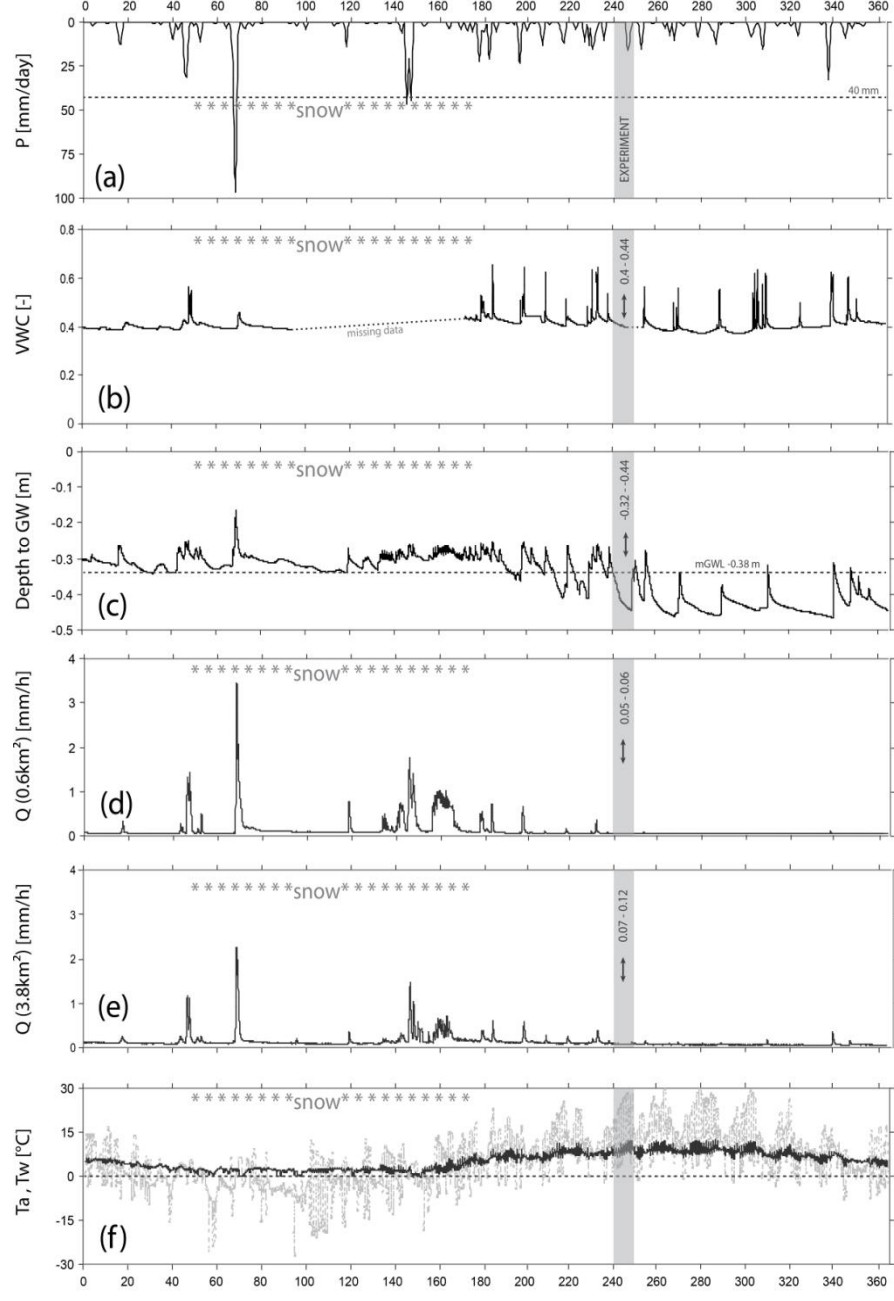

**Fig. 2.** Hydro-meteorological data of the water year 2015 (01.11.2014 – 31.10.2015). **(a) P:** Precipitation [mm d$^{-1}$]. **(b) VWC:** Volumetric water content at hillslope PZ in 0.2 m depth [-]. **(c)** Depth to groundwater level at slope PB3. **(d) Q (0.6 km²):** Discharge [mm h$^{-1}$] of Rokytka headwater. **(e) Q (3.8 km²):** Discharge [mm h$^{-1}$] of Rokytka catchment (3.8 km²). **(f) T$_a$:** Air temperature [°C] (dashed gray line), **T$_w$:** Stream water temperature [°C] (solid black line). The gray bars mark a 10-day period starting with dye tracer sprinkling in the evening of June 29$^{th}$ (day 240 of the water year) and July 8$^{th}$ (day 249 of the water year).





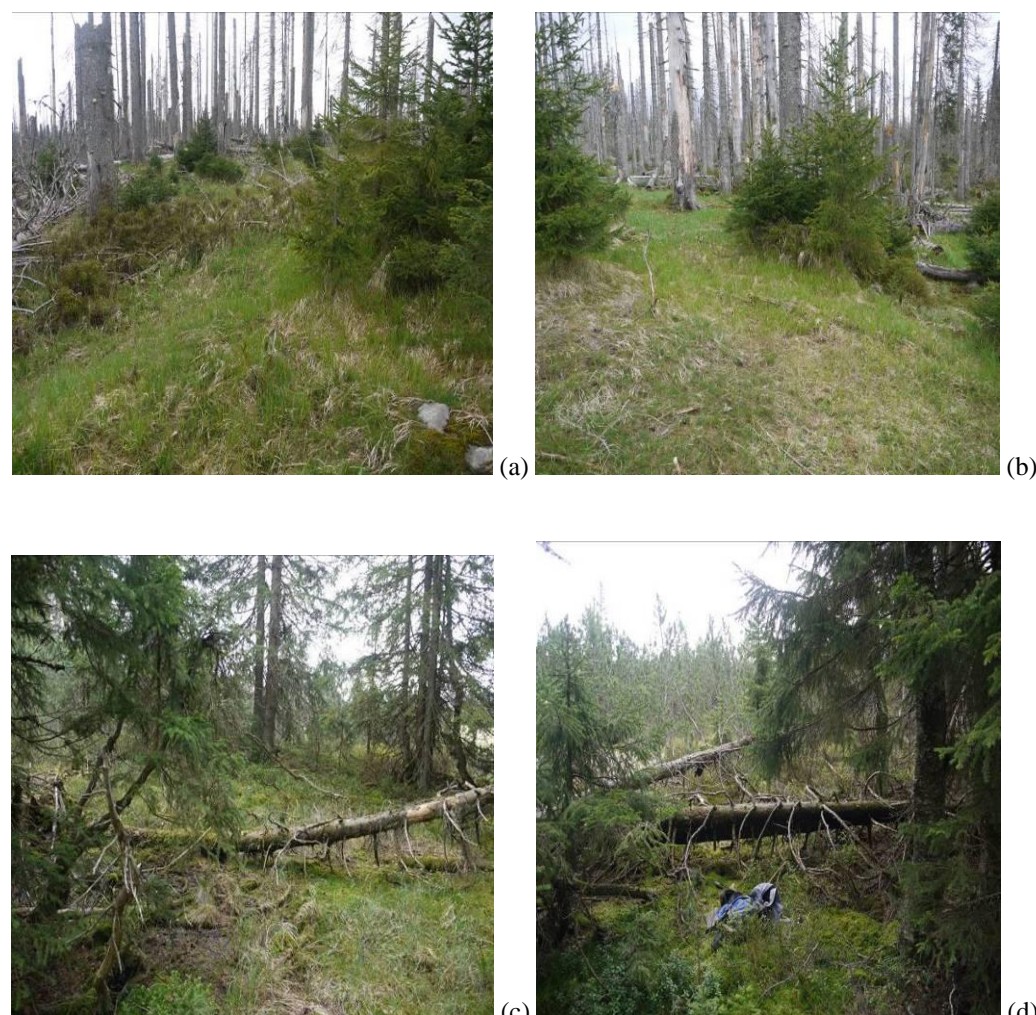

5 **Fig. 3.** Pictures of the application plots of the dye sprinkling experiments at the Rokytka headwater, Šumava Mts., Czech Republic. **(a and b)** Sprinkling plots at the mineral soil hillslope Podzol (PZ2). **(a)** Brilliant Blue plot (BB); **(b)** Sodium-Fluorescein plot (FLC). **(c and d)** Sprinkling plots at the organic soil hillslope Peat Bog (PB3). **(c)** Brilliant Blue plot (BB); **(d)** Sodium-Fluorescein plot (FLC).





**Table 2.** Selected dye sprinkling experiments sorted by land use and scale of dye application.

<table>
<tr><td></td><td>**Land use**</td><td>**Plot studies**</td><td>**Hillslope studies**</td></tr>
<tr><td rowspan="2">**Brilliant Blue**</td><td>**Forest studies**</td><td>Bachmair et al., 2009; Wienhöfer et al., 2009</td><td>Anderson et al., 2009; Wienhöfer et al., 2009</td></tr>
<tr><td>**Grassland studies**</td><td>Weiler and Flühler, 2004; Bachmair et al., 2009; Schneider et al., 2014</td><td>-</td></tr>
<tr><td rowspan="2">**Fluorescein**</td><td>**Forest studies**</td><td>Gerke et al., 2008, 2013, 2015</td><td>Wienhöfer et al., 2009; Weiler and Naef, 2003</td></tr>
<tr><td>**Grassland studies**</td><td>-</td><td>Weiler and Naef, 2003; Schneider, 2007; Schneider et al., 2014</td></tr>
</table>

35





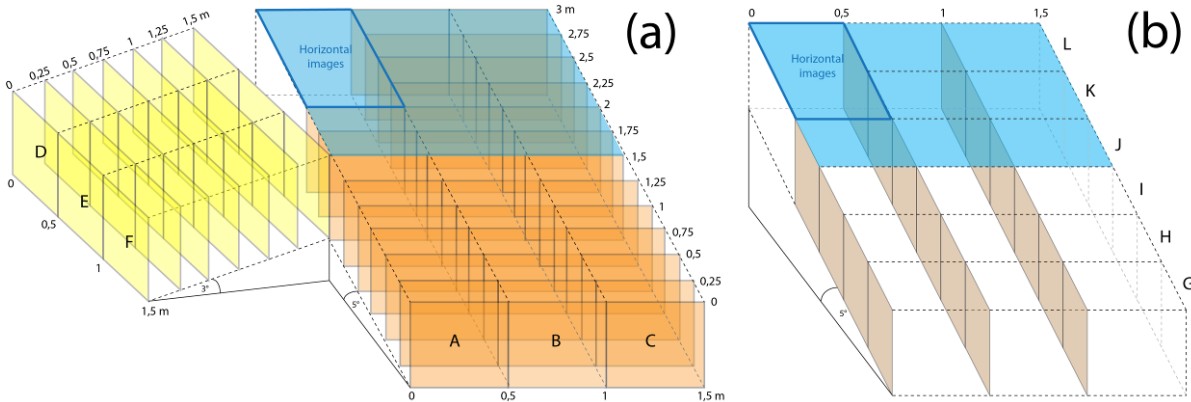

**Fig. 4.** Sketch of the dye tracer experiment excavation directed along the flow lines. **(a)** Scheme of excavation and photography of frontal
soil profiles (orange and yellow sections) up the slope ('columns' A – F). The yellow inclined part (columns D – F) was only excavated
and photographed at plot PB3 to capture the dominant lateral-horizontal preferential flowpaths. **(b)** Scheme of excavation and photography
of lateral soil profiles (gray) parallel to the lateral-horizontal preferential flowpaths (rows G – L). Blue areas show the dye sprinkling plots,
where 'platform' images where taken of horizontal layers of the soil (these layers have the same orientation as the soil layer boundaries).





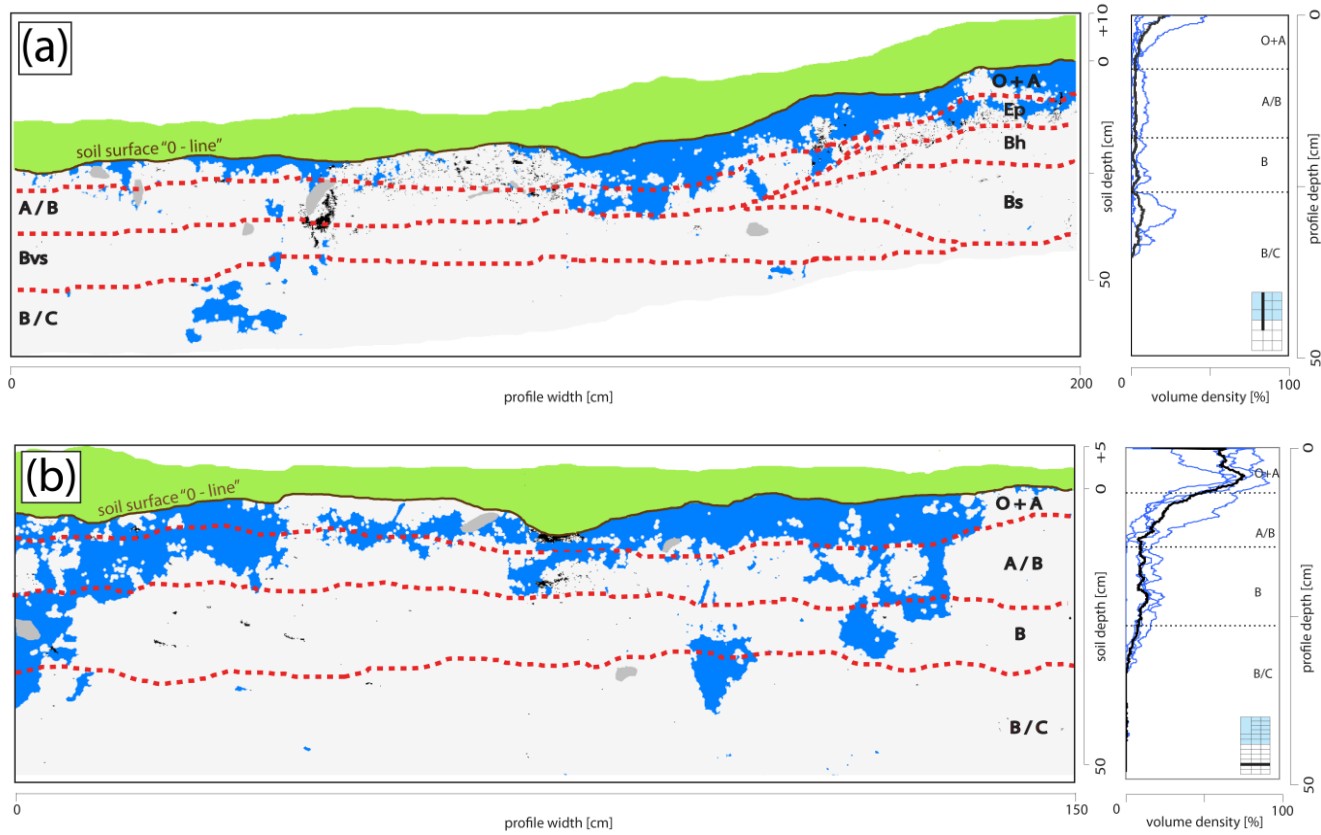

**Fig. 5. (a)** Lateral profile IL1 and **(b)** frontal profile AC0.5 at the Brilliant Blue (BB) sprinkling plot PZ2 at the mineral soil hillslope

5  (Podzol). Blue: BB dye; gray: stones, roots; green: vegetation; black: unclassified shadows, roots; red-dotted line: soil horizon divide. The

charts on the right represent the distribution of the volume density of BB in different soil depths.





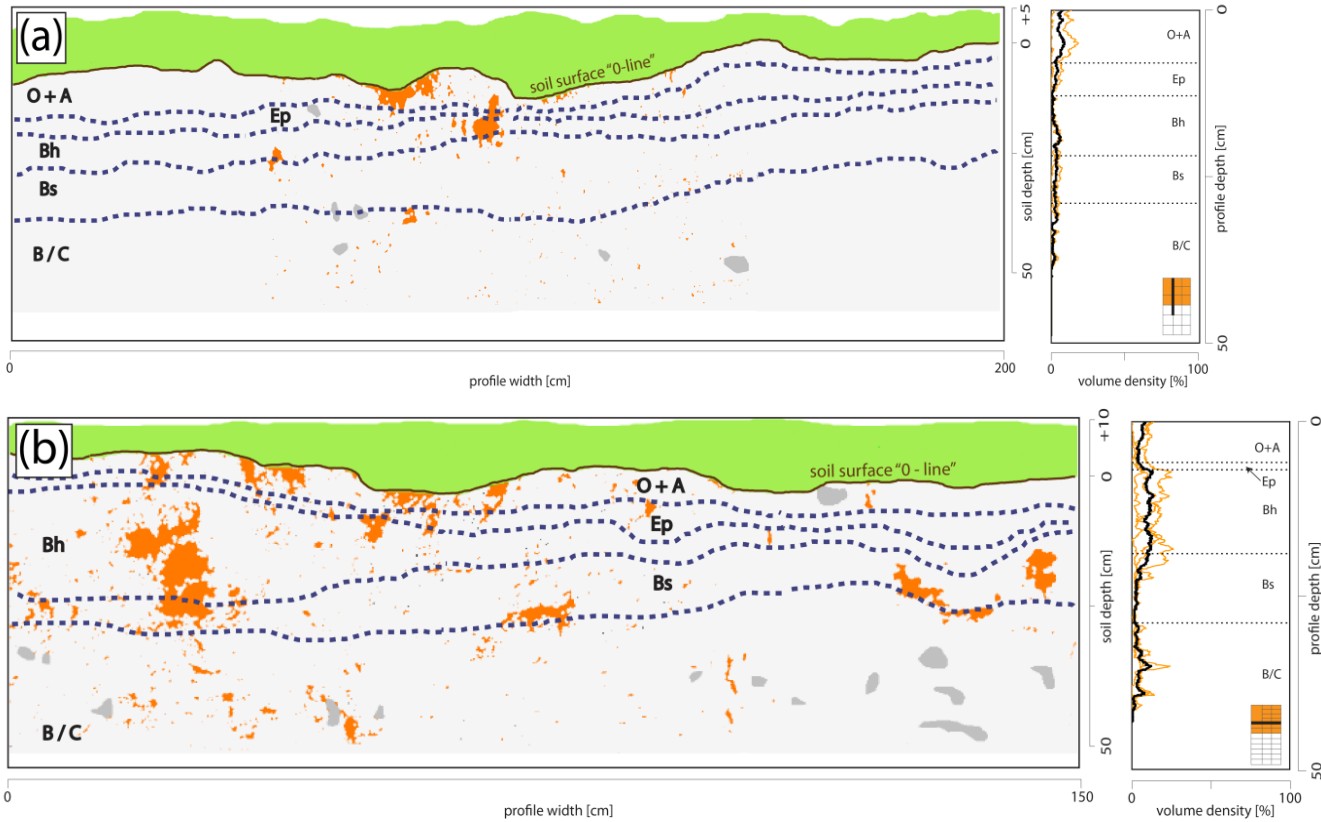

**Fig 6. (a)** Lateral soil profile IL1 and **(b)** frontal soil profile AC0.5 at the Fluorescein (FLC) sprinkling plot PZ2 at the mineral soil
hillslope (Podzol). Orange: FLC dye; gray: stones, roots; green: vegetation; black: unclassified shadows; dark-blue dotted lines: soil
horizon divide. The charts on the right represent the dye volume in different depths below the soil surface.




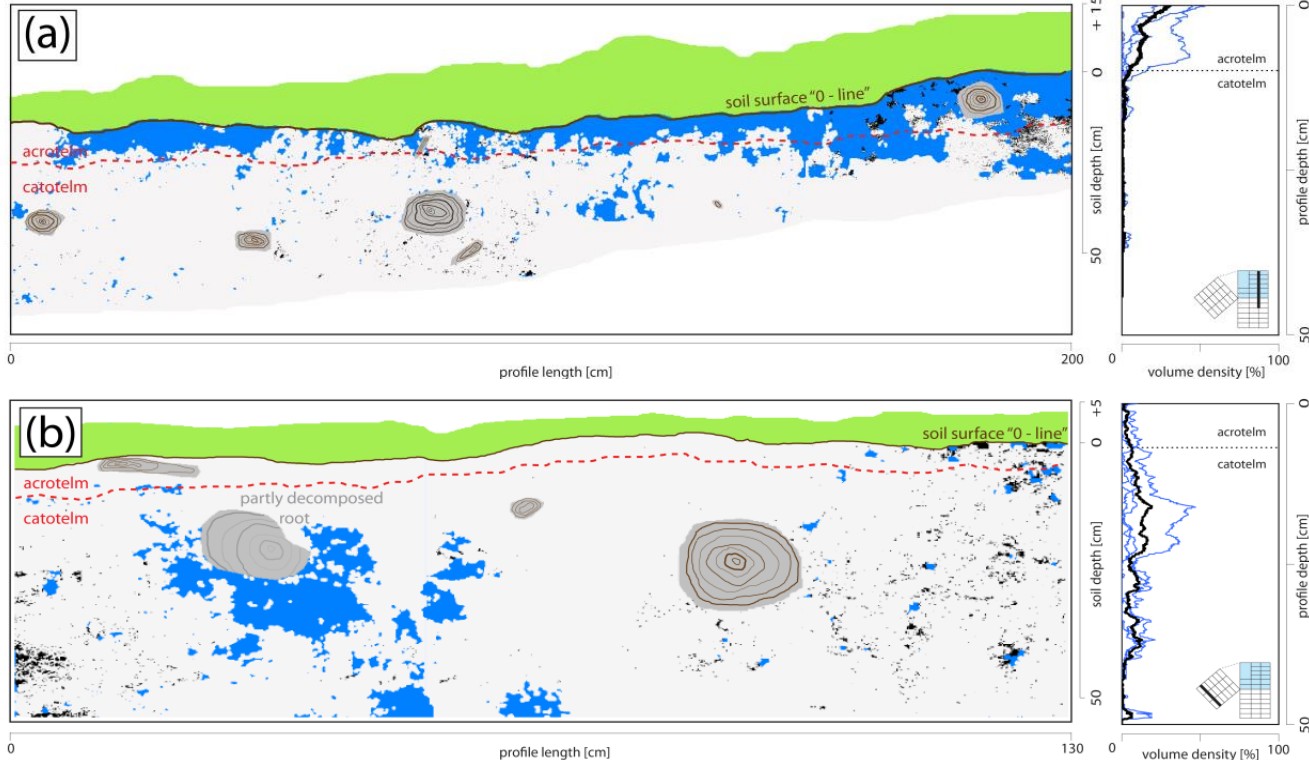

**Fig. 7. (a)** Lateral soil profile IL0.5, and **(b)** frontal soil profile FD0.25 at the Brilliant Blue (BB) sprinkling plot PB3 at the organic soil hillslope (Peat Bog). Blue: BB dye; gray: roots; green: vegetation; black: unclassified shadows; red-dotted line: soil horizon divide. The charts on the right represent the vertical distribution of the volume density of the BB. The reduced width of the soil profiles shown in Fig. 7b and 7c are on the right side of these profiles due to the presence of a tree stump.





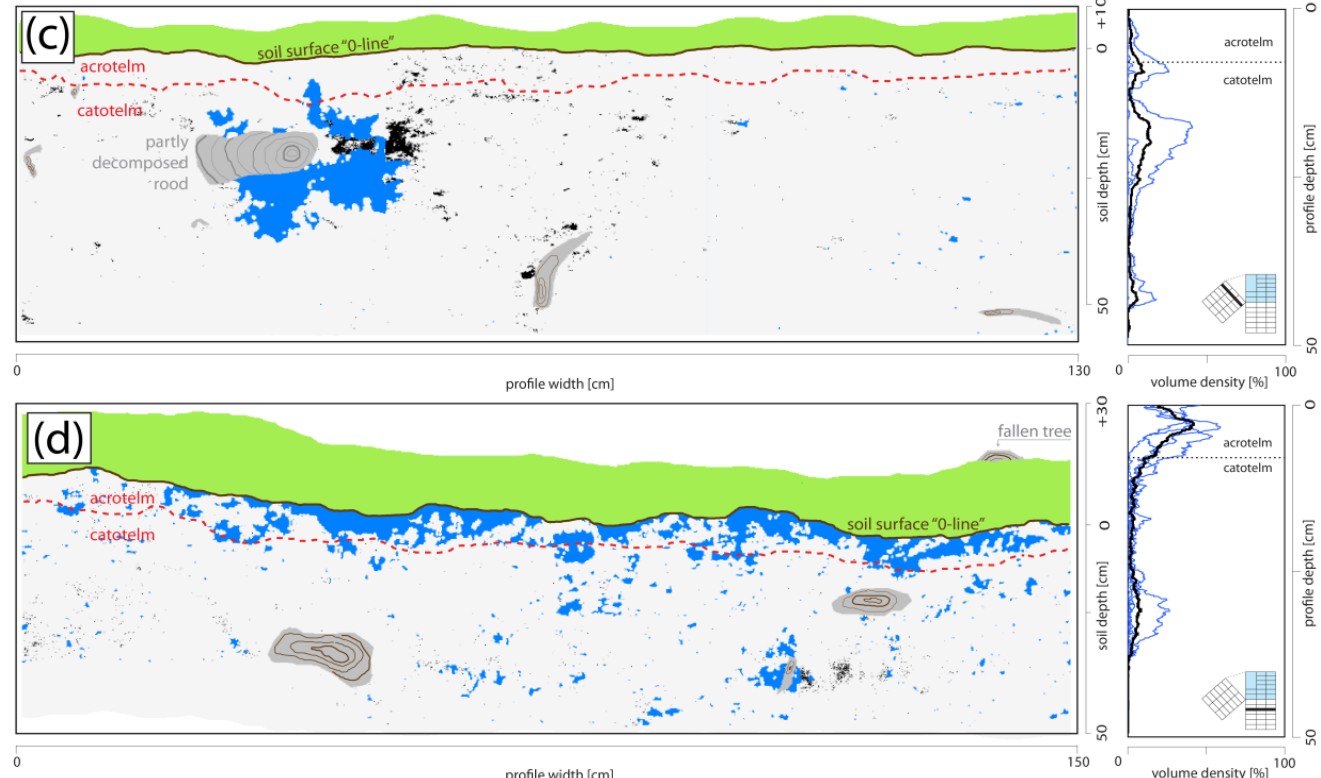

**Fig. 7. (c)** FD1.75 of the NW-branch (left), and **(d)** frontal soil profile AC1 of the SW-branch (right) at the Brilliant Blue (BB) sprinkling plot PB3 at the organic soil hillslope (Peat Bog). Blue: BB dye; gray: roots; green: vegetation; black: unclassified shadows; red-dotted line: soil horizon divide. The charts on the right represent the vertical distribution of the volume density of the BB. The reduced width of the soil profiles shown in Fig. 7b and 7c are due to the presence of a tree stump on the right side of these profiles.




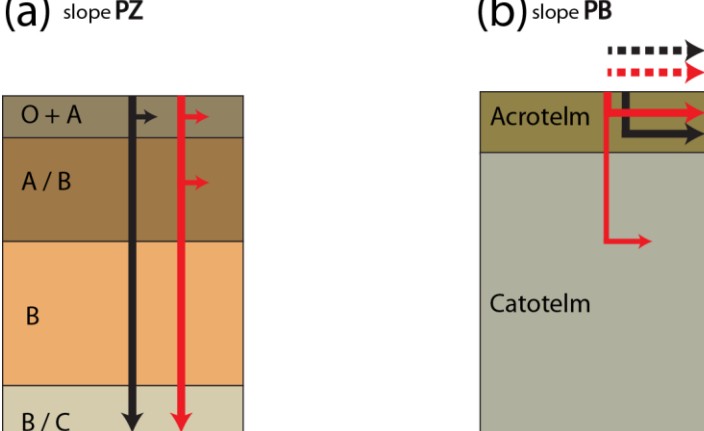

**Fig. 8.** Conceptual models of runoff formation and (subsurface) stormflow for the organic soil hillslope (Peat Bog PB3) and the mineral soil hillslope PZ (Podzol) derived by parallel plot sprinkling experiments with the conservative dye Sodium-Fluorescein (FLC) and the sportive dye Brilliant Blue (BB) at the Rokytka headwater. Detected subsurface flow components at the PZ hillslope **(a)**: Biomat flow, shallow lateral subsurface flow and mostly deep percolation (vertical); and at the organic soil hillslope PB (Peat Bog) **(b)**: Biomat flow at short distances and mostly lateral pipeflow following decayed tree-root systems with long lateral subsurface flow distances. The dashed arrows represent surface flow (saturation overland flow, SOF), which could not be detected during the experiments but has been observed on-site during natural storm events. Black arrows: Hypothesized runoff formation processes according to the *Hydrology of Soil Types* classification (HOST; Boorman et al., 1995); Red arrows: Results of the Šumava experiments.

