# Peer review of "Identification of runoff formation with two dyes in a mid-latitude mountain headwater"

_Hydrology and Earth System Sciences, 2017_

## Referee Comment (RC1) · B. Kohl (Referee) · 17 Mar 2017

Bernhard Kohl

Field testing on runoff formation is highly valuable to get a better understanding of the decisive processes and therefor the topic of the study to investigate different hillslopes with dyes is an interesting question. The authors present an extensive work and an exciting experiment.

The text is organised in a coherent and readable way.

The literature review is modest, but suitable according to the topic. In general literature references need to be revised and should be brought to a uniform condition.

A contradiction occurs within the discussion section concerning the dominant runoff formation processes at the Podzol hillslope. This incongruity calls for an explanation.

However, in my opinion some minor revisions should be done, the authors should be invited to resubmit a revised manuscript.

General comments:

•        Ambiguities concerning runoff processes HOF and SOF need to be clarified.

•        At least two citations mentioned within the text cannot be found in the references-list.

•        Perhaps it is recommended to check the article by a native speaker once more.

Specific comments:

•        literature references need to be revised and should be brought to a uniform condition. E.g. (Hladný and Kašpárek, 2005; Hladný et al., 2005; Flury et al, 1995; Flury and Flühler, 1995; Hümann et al., 201; Hümann et al., 2011…)

•        Perhaps it would be recommended to change the order of the tables 1 and 2 in the chronological appearance

•         (Eriophorum sp. L.); (Sphagnum sp. L.)

•        *"Vertical hydrological conductivity (HCv) was measured on-site with a single-**ring infiltrometer** (Flow-Group Comp.). …. A low HCv in the topsoil is supposed to generate rather surface flow – **likely saturation overland flow (SOF) and possibly Hortonian overland flow (HOF) to a minor extent** – or near-surface biomat flow (BMF; Sidle, 2007) during high intensity storms."*
This line of argument is not understandable: The very low conductivity at the surface (infiltrometer) implies inhibited infiltration and thus infiltration excess = HOF!

•        **"Due to previous rainfall events**, *the soil moisture ranged between 0.40-0.45 VWC."* 0.40 (lowest value) due to previous rainfall events?

•        In order to facilitate the readability of the text perhaps it would be better to change the order of fig.4 ab and 5 ab: ; 4a frontal, 4 b lateral; 5a frontal 5b lateral

•        *"This is noteworthy since the proportion of the Peat Bog ranges from **60%** at the 2nd order stream headwater to less than 30% at the 3rd order 15 stream catchment; the remaining areas are covered by Podzol."* Table 1 gives PB 44% RH. 60% or 44%?

•        *"According to the runoff formation decision scheme by Scherrer and Naef (2003), the dominant runoff formation process at the Podzol hillslope **can be classified as a combination of delayed Hortonian overland flow (HOF) and delayed subsurface stormflow (SSF2)**."*  This is a clear contradiction to the observed results. At P16 L12 ist is mentioned that "…deep percolation into the bedrock dominated…". So deep percolation DP is the primary runoff formation process at the Podzol hillslope (fig.8). This incongruity calls for an explanation.

•        *"Our hypothesis of HOF was confirmed for the Peat Bog hillslope…"*   see the above statements on HOF, SOF

•        Fig.1     Well designed illustration. WLPWS* What does the asterisk stand for?

•        Fig.8b   (*saturation overland flow, SOF)* see the above statements on HOF, SOF

Technical corrections:

P8 L9    for two weeks.

P11 L7    to test

P13 L6  Burt

---

## Author Comment (AC1) · 6 Apr 2017

First, we would like to thank the referee for his review and the valuable comments. Typos and ambiguities in references were corrected and the manuscript was revised accordingly.

General comments:

1. Possible ambiguities concerning the runoff process HOF and SOF have been clarified (see point 4, 8, 9 in specific comments).

2. Agreed and corrected.

3. Our manuscript has been checked by a proficient native speaker.

Specific comments:

[Figure]

1. Agreed and corrected.

2. Agreed and order of tables changed accordingly.

3. Agreed and changed.

4. We agree, there is a contradiction in our data/field observations, however in our opinion none of the data/observations overrules the other. Vertical hydrological conductivity (HCv) was measured in situ with a single-ring infiltrometer. The measured HCv was very low, on the other hand no surface flow was observed even during heavy rainfall events (field observation and thus "soft data"). The very low HCv measured in the soil matrix suggests – but does not prove – 'Hortonian overland flow' (HOF) at the hillslope, as macropores for instance are likely not well represented in this small-scale on-site measurement (scale issue). As the scale of our HCv measurement is in the order of decimeters (10 cm x 10 cm), the focus is on measuring the hydraulic conductivity of the soil matrix. Thus, the soil matrix's vertical hydraulic conductivity properties are not the ultimate answer on the prevailing runoff formation processes at our test site – it's rather one aspect contributing to the larger picture. As a matter of fact, the contradiction between the soil matrix properties (HCv) and the field observations were the motivation to investigate the runoff formation processes at the hillslopes with dye experiments. We understand the differences between the terms 'Hortonian Overland Flow' (HOF) and 'Saturation Overland Flow' (SOF), definitions are given in subsection 8.

5. Soil moisture was measured on-site. Previous rainfall events were recorded off-site (nearby meteo station in 400 m distance) several days before our experiment started, see Fig. 2. The Peat Bog stays in a humid climate permanently wet, thus 40% VWC is for the Rokytka test site as dry as it gets in early summer. The summer 2015 was relatively dry and the previous rainfall events were rather small and less frequent than in a normal year. Text will be reformulated accordingly.

6. Change order of Fig. 4: Agreed, figure modified accordingly.

7. Table 1 (changed order as suggested by the referee: now Table 2) shows the coverage of the soil type 'Histosol' (WRB, 2006). 'Histosols' are the dominant soil type in Peat Bog areas, but other soils types occur as well: partly 'histic soils' and partly 'Fluvisols' (WRB, 2006) at the bottom of the valley. Therefore, the given values are correct: Rokytka Headwater (RH) = 44% and Otava River Headwater (OR) = 20%.

8. We agree with the referee, that the dominant runoff formation mechanism detected by our dye experiments at the Podzol hillslope was clearly 'deep percolation' (DP). The experimental results do not suggest that HOF or SOF are dominant runoff formation processes for mid-range intensity storms. However, 'surface-near flowpaths' cannot be completely excluded concerning larger storms with higher intensities – we observed slightly detectable, short-distance initiations of surface-near flowpaths (yet not a dominant feature at our experimental esttings). The point is that in the discussion we intend to confront our experimental findings at the Rokytka test site with an established runoff formation classification scheme (Scherrer and Naef, 2003) for flood-causing extreme events, which can be applied to one of our hillslopes. This scheme suggests delayed HOF and delayed SSF for the Podzol hillslope. The Scherrer scheme has been developed and applied for high-intensity storm events in Switzerland (intensities > 50mm/h). Our experimental intensities represent rather average annual storms (intensities 20-30 mm/h). We will clarify that in the final version of the manuscript. The Scherrer's scheme is limited to sites without shallow groundwater, thus it can be applied to the Podzol hillslope (PZ) only.

9. Agreed and will be clarified in the final manuscript. We will point more clearly to the observed main subsurface drainage feature.

10. Fig. 1 The asterisk is now explained in the caption of the figure.

11. Fig. 8b See above (-> points 8, 9 and 4).

12. Technical corrections: Agreed and changed as suggested.

References:

IUSS Working Group WRB.: World reference base for soil resources 2006. 2nd edition. World Soil Resources Reports No. 103. FAO, Rome. ISBN 92-5-105511-4, 2006.

Scherrer, S. and Naef, F.: A decision scheme to indicate dominant hydrological flow processes on temperate grassland, Hydrol. Process., 17(2), 391 − 401, doi:10.1002/hyp.1131, 2003.

---

## Referee Comment (RC2) · Anonymous Referee #2 · 7 Apr 2017

Summary In the manuscript by Lukáš Vlček et al. runoff formation processes at two hillslopes of a headwater catchment in Czech Republic have been investigated. Sprinkling experiments with two different dye tracers were performed at a Podzol site and a Peat Bog site each. Lateral and frontal profiles were excavated, photographed and stained areas were detected by image analysis. The results showed that biomat flow through the upper, organic litter layer and lateral preferential flow along dead trees and roots prevailed at the Peat Bog site. Vertical percolation was the dominant process at the Podzol site, although the presence of lateral preferential flow could be proven as well.

General comments The study presented in the manuscript is interesting and relevant. The manuscript is well written, the English expression is very good and the readability too but it is rather long for a research paper. The manuscript should be shortened

significantly to be more concise (beyond the suggestions in the specific comments). My overall impression is that the work deserves to be published. I recommend publication in HESS after minor revisions.

Specific comments P3 L12-22: The three points mentioned are comprehensible and interesting descriptions of the two experimental sites. It is, however, not totally clear to me why they are reasons or advantages for selecting the two sites for the experiment (as stated in P3 L12). P4 L4-13: In the first and third points you stated that you also tried to estimate/quantify infiltration, preferential flow and vertical percolation. I suggest to delete the words "estimate/quantify" here, because you do not provide numbers that really quantify these fluxes. P5 L28 – P6 L4: The hydrological metrics MQ, MHQ, MNQ and HQ indicate flow rates and, thus, they have the dimension $L^3 T^{-1}$ (e.g. the unit litre per second). In the context of the manuscript it is meaningful to relate these metrics to the catchment area (Dimension: $L^1 T^{-1}$, unit e.g. mm per hour), but they should not be addressed as fluxes and the acronyms MQ, MHQ, MNQ and HQ should not be used. Maybe you can call it "discharge per area" or "unit discharge". P6 L12: Please add information about the concentration of tracer solution or the total mass of tracer applied. P6 L29 – P7 L11: I understood that you took both frontal and lateral profiles from each irrigated plot, right? For me it is difficult to imagine how the frontal profiles (Fig. 4a) could be excavated first without destroying the lateral profiles (Fig. 4b). P7 L30 – P8 L3: When only an area of 1.5m*1.5m=2.25m2 is irrigated, the experimental conditions are of course different to a real rain event where the hillslope receives much more water. Thus, we cannot expect that the flow patterns detected in the experiment are simply transferable to real rain events. I am aware that this is not a very innovative comment since we all have to deal with such issues when performing field experiments. However, I think it should be mentioned and discussed in the discussion section. P8 L17: This sentence can be omitted because the content is already mentioned in the method section. I had problems to find the positions of specific profiles (mentioned in the text and shown in the Figures) in Figure 4:   Please make the reader early aware of the small sketches in the lower right corners of Fig. 5-7 indicating the positions of

individual profiles. They are very helpful but it was too late that I took notice of them. • Figure 4: A north arrow in Fig. 4 might help to orientate when directions are mentioned in the text. • Please check if the designations of single profiles do always correspond to Figure 4. For example: I cannot find the location of the profile FD1.75 (Fig 7c) in Figure 4. Taking the small sketch in Fig. 7c into account I would guess that profile FD1.25 is shown In Fig. 7c. • P8 L22-23: Please mention already here that Fig. 5b shows a profile outside (downstream) the sprinkling plot. That information can then be omitted from P8 L30. • P9 L18-20: Is 10.5 m downslope correct? Was the distance between frontal profiles really 0.5m? From the yellow section in Fig. 4a I would expect 1.5 m downslope and distances of 0.25 m. Please check this. P10 L15-23: It is an important result that no FLC has been found in springs and in the stream. However, this paragraph contains much methodological information that should be shifted to the Methods section. P10 L25 – P11 L5: I agree with the content of the section. However, most of it is already mentioned in the Introduction section and can be omitted here. P13 L32 – P14 L3: This issue is already mentioned in the previous paragraph. P14 L3-6: Is the colour of Fluorescin that can be detected during daylight with the human eyes also affected by pH? If not, you could not conclude in P14 L5/6 that the dye has not been "affected by pH changes". I have always thought before that only the fluorescence is pH dependent, but not the colour. However, I am not sure about this point. P14 L8-28: Is it also possible that the tracer was very strongly diluted before it reached the stream? If yes, it would also be possible that the tracer concentration in the stream was below the detection limit of your analytical method. Could you provide a rough estimation of dilution at the slope? Maybe you are then able to remove my concerns. This comment is related to my comment on P6 L12. P15 L18 – P16 L18: The conclusion should be shorter and more concise: • Issues that have exhaustively been discussed before should not be repeated again, e.g. the discussion about a pH effect. • It would be nice to have a few clear and concise statement about what we can learn from this study. • Point out a few take home messages related to the four specific points that have been defined on page 4 as the objectives of the study. Table 1: What is the information

about the Otava River used for in this study? Figure 4: In both sketches positions of horizontal images are indicated. Is that information needed for this manuscript? The horizontal images are not mentioned in the text. Figures 5 – 7, lateral profiles: Please insert vertical lines indicating boundaries of the sprinkling plot.

Technical corrections Table1 & 2: Switch the positions of Table 1 and Table 2. Currently Table 2 is mentioned first in the text. P5 L16: The word "both" should be deleted from the sentence. P15 L27-28: Use past tense.

Please also note the supplement to this comment:
http://www.hydrol-earth-syst-sci-discuss.net/hess-2017-77/hess-2017-77-RC2-supplement.pdf

---

## Author Comment (AC2) · 2 May 2017

We would like to thank the referee for his review and the helpful comments.

Responses to comments: 1. Agreed.

2. Agreed, it will be clarified.

3. Agreed, it will be deleted.

4. Agreed, it will be clarified.

5. Agreed, it will be changed.

6. Profiles were excavated stepwise in 0.5 m-wide segments. The pictures were first taken in squares of 0.5 m x 0.5 m of a profile section, and later processed, analyzed,

and joined in the software and finally displayed in our figures in 1.5m-wide profiles.

7. We agree to include the issue in the discussion. However, we believe that the detected preferential flow patterns give valuable insights into active preferential flowpaths during natural rain events. The flow patterns are likely intensified and additional flow processes might occur, but our experimentally detected dominant flowpaths will very likely play a crucial role. Moreover, we applied our experiments in a dimension [1.5 m-scale] which is well established in the dye experiment literature.

8. Agreed, it will be changed.

9. Agreed, text will be changed accordingly.

10. Agreed, sentence will be clarified.

11. Agreeed, it will be changed.

12. Agreed, text will be shortened.

13. Agreed, text will be shortened.

14. This statement will be clarified. Background info: The pH value slightly influences the visible (= "eyeball-detectable") color of FLC solutions, as it changes from neon greenish-yellow to a dull goldish-yellow depending on pH. This is because when FLC is in its ionized (= deprotonated) form at pH > ~6, it has significantly different fluorescence behavior/characteristics than when it is at or below pH ~6 (below pH ~6 it is mostly protonated, H+ group adds to COO–). The fluorescence of FLC is due to the photo-physical characteristics of the fluorescein anion at pH > ~6. Its color (visible absorbance) is influenced by its concentration. The most intense neon greenish-yellow color (visible absorbance) occurs at concentrations of 0.1 to 1 g L-1. The most intense fluorescence occurs when the salt Sodium-Fluorescein (= Uranine) is fully dissociated, according to Käss (1998). Absorbance and fluorescence are both observed in the same spectral range (~450-550 nm).

15. This point is answered together with point 14.

16. The high sorption of FLC to organic matter together with the pH likely explain the suppression the fluorescence. Also, sprinkling of the plots only (and not the entire hillslope), may have prevented observation of FLC in the stream because of lack of hydrological connectivity by lower saturation of the soils outside of the sprinkling plots. Since we observed that Brilliant Blue was not strongly diluted at the Peat Bog hillslope (still strong blue color near the stream), we can assume that dilution was rather small. Our experimental data is not sufficient for a detailed calculation, as BB concentrations are well suited for qualitative detection but not for detailed quantification (as it is possible for FLC).

17. P15 Agreed, text will be shortened.

18. This table shows a comparison of the headwaters of the upper Otava River – which consists of many small headwaters and forms one of the most prominent catchments in the Šumava Mts. region. Our experimental catchment "Rokytka Headwater" is a sub-catchment of the upper Otava River. The Otava River catchment is the reference & target area, e.g for scaling up of runoff formation processes and for the implementation of flood protection measures etc.

19. Agreed, will be removed.

20. Agreed, information will be added.

21. Agreed, will be changed.

22. Agreed.

23. Agreed, will be changed.

---

## Author Comment (AC3) · 2 May 2017

We would like to thank the referee for his review and the helpful comments.

General comments: The manuscript should be shortened C1 HESSD Interactive comment Printer-friendly version Discussion paper significantly to be more concise (beyond the suggestions in the specific comments). My overall impression is that the work deserves to be published. I recommend publication in HESS after minor revisions.

Authors agree, text will be shorten.

Specific comments P3 L12-22: The three points mentioned are comprehensible and interesting descriptions of the two experimental sites. It is, however, not totally clear to me why they are reasons or advantages for selecting the two sites for the experiment

(as stated in P3 L12).

Authors agree, it will be clarified.

P4 L4-13: In the first and third points you stated that you also tried to estimate/quantify infiltration, preferential flow and vertical percolation. I suggest to delete the words "estimate/quantify" here, because you do not provide numbers that really quantify these fluxes.

Autrhors agree, it will be deleted.

P5 L28 – P6 L4: The hydrological metrics MQ, MHQ, MNQ and HQ indicate flow rates and, thus, they have the dimension L3 T-1 (e.g. the unit litre per second). In the context of the manuscript it is meaningful to relate these metrics to the catchment area (Dimension: L1 T-1, unit e.g. mm per hour), but they should not be addressed as fluxes and the acronyms MQ, MHQ, MNQ and HQ should not be used. Maybe you can call it "discharge per area" or "unit discharge".

Authors agree, it will be clarified.

P6 L12: Please add information about the concentration of tracer solution or the total mass of tracer applied.

Authors agree, information will be changed.

P6 L29 – P7 L11: I understood that you took both frontal and lateral profiles from each irrigated plot, right? For me it is difficult to imagine how the frontal profiles (Fig. 4a) could be excavated first without destroying the lateral profiles (Fig. 4b).

Authors: Profiles were excavated stepwise in 0.5 m-wide segments. The pictures were first taken in squares of 0.5 m x 0.5 m of a profile section, and later processed, analyzed, and joined in the software and finally displayed in our figures in 1.5m-wide profiles.

P7 L30 – P8 L3: When only an area of 1.5m*1.5m=2.25m2 is irrigated, the experimental conditions are of course different to a real rain event where the hillslope receives much more water. Thus, we cannot expect that the flow patterns detected in the experiment are simply transferable to real rain events. I am aware that this is not a very innovative comment since we all have to deal with such issues when performing field experiments. However, I think it should be mentioned and discussed in the discussion section.

Authors: We agree to include the issue in the discussion. However, we believe that the detected preferential flow patterns give valuable insights into active preferential flowpaths during natural rain events. The flow patterns are likely intensified and additional flow processes might occur, but our experimentally detected dominant flowpaths will very likely play a crucial role. Moreover, we applied our experiments in a dimension [1.5 m-scale] which is well established in the dye experiment literature.

P8 L17: This sentence can be omitted because the content is already mentioned in the method section. I had problems to find the positions of specific profiles (mentioned in the text and shown in the Figures) in Figure 4: âAËŸ c Please make the reader early aware ′ of the small sketches in the lower right corners of Fig. 5-7 indicating the positions of C2 HESSD Interactive comment Printer-friendly version Discussion paper individual profiles. They are very helpful but it was too late that I took notice of them. âAËŸ c Figure 4: A north arrow in Fig. 4 might help to orientate when directions are ′ mentioned in the text. âAËŸ c Please check if the designations of single profiles do always ′ correspond to Figure 4. For example: I cannot find the location of the profile FD1.75 (Fig 7c) in Figure 4. Taking the small sketch in Fig. 7c into account I would guess that profile FD1.25 is shown In Fig. 7c. âAËŸ c P8

Authors agree, it will be changed.

L22-23: Please mention already here that ′ Fig. 5b shows a profile outside (downstream) the sprinkling plot. That information can then be omitted from P8 L30. âAËŸ c P9 L18-20: Is 10.5 m downslope correct? Was the ′ distance between frontal profiles

really 0.5m? From the yellow section in Fig. 4a I would expect 1.5 m downslope and distances of 0.25 m. Please check this.

Authors agree, text will be changed accordingly.

P10 L15-23: It is an important result that no FLC has been found in springs and in the stream. However, this paragraph contains much methodological information that should be shifted to the Methods section. Authors agree, it will be changed. P10 L25 – P11 L5: I agree with the content of the section. However, most of it is already mentioned in the Introduction section and can be omitted here. Authors agree, text will be shortened.

P13 L32 – P14 L3: This issue is already mentioned in the previous paragraph.

Authors agree, text will be shortened.

P14 L3-6: Is the colour of Fluorescin that can be detected during daylight with the human eyes also affected by pH? If not, you could not conclude in P14 L5/6 that the dye has not been "affected by pH changes". I have always thought before that only the fluorescence is pH dependent, but not the colour. However, I am not sure about this point.

Authors: This statement will be clarified. Background info: The pH value slightly influences the visible (= "eyeball-detectable") color of FLC solutions, as it changes from neon greenish-yellow to a dull goldish-yellow depending on pH. This is because when FLC is in its ionized (= deprotonated) form at pH > ∼6, it has significantly different fluorescence behavior/characteristics than when it is at or below pH ∼6 (below pH ∼6 it is mostly protonated, H+ group adds to COO–). The fluorescence of FLC is due to the photo-physical characteristics of the fluorescein anion at pH > ∼6. Its color (visible absorbance) is influenced by its concentration. The most intense neon greenish-yellow color (visible absorbance) occurs at concentrations of 0.1 to 1 g L-1. The most intense fluorescence occurs when the salt Sodium-Fluorescein (= Uranine) is fully dissociated,

according to Käss (1998). Absorbance and fluorescence are both observed in the same spectral range (∼450-550 nm).

P14 L8-28: Is it also possible that the tracer was very strongly diluted before it reached the stream? If yes, it would also be possible that the tracer concentration in the stream was below the detection limit of your analytical method. Could you provide a rough estimation of dilution at the slope? Maybe you are then able to remove my concerns. This comment is related to my comment on P6 L12.

Authors: This point is answered together with point 14.

P15 L18 – P16 L18: The conclusion should be shorter and more concise: âAËŸ c Issues that have exhaustively been discussed before ′ should not be repeated again, e.g. the discussion about a pH effect. âAËŸ c It would be ′ nice to have a few clear and concise statement about what we can learn from this study. âAËŸ c Point out a few take home messages related to the four specific points that have ′ been defined on page 4 as the objectives of the study.

Authors agree, text will be modified and shortened.

Table 1: What is the information C3 HESSD Interactive comment Printer-friendly version Discussion paper about the Otava River used for in this study?

Authors: This table shows a comparison of the headwaters of the upper Otava River – which consists of many small headwaters and forms one of the most prominent catchments in the Šumava Mts. region. Our experimental catchment "Rokytka Headwater" is a sub-catchment of the upper Otava River. The Otava River catchment is the reference & target area, e.g for scaling up of runoff formation processes and for the implementation of flood protection measures etc.

Figure 4: In both sketches positions of horizontal images are indicated. Is that information needed for this manuscript? The horizontal images are not mentioned in the text.

[Figure]

Authors agreed, information about horizontal images will be removed.

Figures 5 – 7, lateral profiles: Please insert vertical lines indicating boundaries of the sprinkling plot. Technical corrections Table1 & 2: Switch the positions of Table 1 and Table 2. Currently Table 2 is mentioned first in the text.

Authors agreed, will be changed.

P5 L16: The word "both" should be deleted from the sentence.

Authors agree.

P15 L27-28: Use past tense.

Authors agree.

---

## Author Comment (AC4) · 2 May 2017

First, we would like to thank the referee for his review and the valuable comments. Typos and ambiguities in references were corrected and the manuscript was revised accordingly

General comments: Ambiguities concerning runoff processes HOF and SOF need to be clarified.

Authors: Possible ambiguities concerning the runoff process HOF and SOF have been clari- fied (see point 4, 8, 9 in specific comments).

At least two citations mentioned within the text cannot be found in the references-list.

Authors agree and it will be corrected.

Perhaps it is recommended to check the article by a native speaker once more.

Authors: Our manuscript has been checked by a proficient native speaker.

Specific comments: 1. literature references need to be revised and should be brought to a uniform condition. E.g. (Hladná and Kašpárek, 2005; Hladná et al., 2005; Flury et al, 1995; Flury and Flühler, 1995; Hümann et al., 201; Hümann et al., 2011...)

Authors agree, it will be corrected.

2. Perhaps it would be recommended to change the order of the tables 1 and 2 in the chronological appearance

Authors agree and order of tables will be changed accordingly.

3. (Eriophorum sp. L.); (Sphagnum sp. L.)

Authors agree and it will be changed.

4. "Vertical hydrological conductivity (HCv) was measured on-site with a single-ring infiltrometer (Flow-Group Comp.). .... A low HCv in the topsoil is supposed to generate rather surface flow – likely saturation overland flow (SOF) and possibly Hortonian overland flow (HOF) to a minor extent – or near-surface biomat flow (BMF; Sidle, 2007) during high intensity storms." This line of argument is not understandable: The very low conductivity at the surface (infiltrometer) implies inhibited infiltration and thus infiltration excess = HOF!

Authors: We agree, there is a contradiction in our data/field observations, however in our opinion none of the data/observations overrules the other. Vertical hydrological conductivity (HCv) was measured in situ with a single-ring infiltrometer. The measured HCv was very low, on the other hand no surface flow was observed even during heavy rainfall events (field observation and thus "soft data"). The very low HCv measured in the soil matrix suggests – but does not prove – 'Hortonian overland flow' (HOF) at the hillslope, as macropores for instance are likely not well represented in this small-scale

on-site measurement (scale issue). As the scale of our HCv measurement is in the order of decimeters (10 cm x 10 cm), the focus is on measuring the hydraulic conductivity of the soil matrix. Thus, the soil matrix's vertical hydraulic conductivity properties are not the ultimate answer on the prevailing runoff formation processes at our test site – it's rather one aspect contributing to the larger picture. As a matter of fact, the contradiction between the soil matrix properties (HCv) and the field observations were the motivation to investigate the runoff formation processes at the hillslopes with dye experiments. We understand the differences between the terms 'Hortonian Overland Flow' (HOF) and 'Saturation Overland Flow' (SOF), definitions are given in subsection 8.

5. "Due to previous rainfall events, the soil moisture ranged between 0.40-0.45 VWC." 0.40 (lowest value) due to previous rainfall events?   In order to facilitate the readability of the text perhaps it would be better to change the order of fig.4 ab and 5 ab: 4a frontal, 4 b lateral; 5a lateral 5b frontal; 4a frontal, 4 b lateral; 5a frontal 5b lateral.

Authors: Soil moisture was measured on-site. Previous rainfall events were recorded off-site (nearby meteo station in 400 m distance) several days before our experiment started, see Fig. 2. The Peat Bog stays in a humid climate permanently wet, thus 40% VWC is for the Rokytka test site as dry as it gets in early summer. The summer 2015 was relatively dry and the previous rainfall events were rather small and less frequent than in a normal year. Text will be reformulated accordingly.

6. Change order of Fig. 4.

Authors agree, figure will be modified accordingly.

7. "This is noteworthy since the proportion of the Peat Bog ranges from 60% at the 2nd order stream headwater to less than 30% at the 3rd order 15 stream catchment; the remaining areas are covered by Podzol." Table 1 gives PB 44% RH. 60% or 44%?

Authors: Table 1 (changed order as suggested by the referee: now Table 2) shows

the coverage of the soil type 'Histosol' (WRB, 2006). 'Histosols' are the dominant soil type in Peat Bog areas, but other soils types occur as well: partly 'histic soils' and partly 'Fluvisols' (WRB, 2006) at the bottom of the valley. Therefore, the given values are correct: Rokytka Headwater (RH) = 44% and Otava River Headwater (OR) = 20%. 8. "According to the runoff formation decision scheme by Scherrer and Naef (2003), the dominant runoff formation process at the Podzol hillslope can be classified as a combination of delayed Hortonian overland flow (HOF) and delayed subsurface stormflow (SSF2)." This is a clear contradiction to the observed results. At P16 L12 ist is mentioned that "...deep percolation into the bedrock dominated...". So deep percolation DP is the primary runoff formation process at the Podzol hillslope (fig.8). This incongruity calls for an explanation.

Authors: We agree with the referee, that the dominant runoff formation mechanism detected by our dye experiments at the Podzol hillslope was clearly 'deep percolation' (DP). The experimental results do not suggest that HOF or SOF are dominant runoff formation processes for mid-range intensity storms. However, 'surface-near flowpaths' cannot be completely excluded concerning larger storms with higher intensities – we observed slightly detectable, short-distance initiations of surface-near flowpaths (yet not a dominant feature at our experimental esttings). The point is that in the discussion we intend to confront our experimental findings at the Rokytka test site with an established runoff formation classification scheme (Scherrer and Naef, 2003) for flood-causing extreme events, which can be applied to one of our hillslopes. This scheme suggests delayed HOF and delayed SSF for the Podzol hillslope. The Scherrer scheme has been developed and applied for high-intensity storm events in Switzerland (intensities > 50mm/h). Our experimental intensities represent rather average annual storms (intensities 20- 30 mm/h). We will clarify that in the final version of the manuscript. The Scherrer0s scheme is limited to sites without shallow groundwater, thus it can be applied to the Podzol hillslope (PZ) only.

9. "Our hypothesis of HOF was confirmed for the Peat Bog hillslope..." see the above

statements on HOF, SOF.

Authors agree and it will be clarified in the final manuscript. We will point more clearly to the observed main subsurface drainage feature.

10. Fig.1 Well designed illustration. WLPWS* What does the asterisk stand for?

Authors: The asterisk is now explained in the caption of the figure.

11. Fig.8b (saturation overland flow, SOF) see the above statements on HOF, SOF.

Authors: See above (-> points 8, 9 and 4)

12. Technical corrections: P8 L9 for two weeks. P11 L7 to to test P13 L6 Burt Burts

Authors agree and it will be changed as suggested.

References: IUSS Working Group WRB.: World reference base for soil resources 2006. 2nd edition. World Soil Resources Reports No. 103. FAO, Rome. ISBN 92-5-105511-4, 2006.

Scherrer, S. and Naef, F.: A decision scheme to indicate dominant hydrological flow processes on temperate grassland, Hydrol. Process., 17(2), 391 – 401, doi:10.1002/hyp.1131, 2003.